# Beyond Pairwise Connections: Extracting High-Order Functional Brain Network Structures under Global Constraints

**Ling Zhan**[1,2], **Junjie Huang**[1], **Xiaoyao Yu**[1], **Wenyu Chen**[1,3], **Tao Jia**[1,2,4*]

[1]College of Computer and Information Science, Southwest University, Chongqing, China

[2]Chongqing Key Laboratory of Brain-Inspired Cognitive Computing
and Educational Rehabilitation for Children with Special Needs,
Chongqing Normal University, Chongqing, China

[3]School of Science, Guangxi University of Science and Technology, Guangxi, China

[4]College of Computer and Information Science, Chongqing Normal University, Chongqing, China

`zl0327@email.swu.edu.cn`, `junjiehuang@swu.edu.cn`, `xiaoyaoyu@email.swu.edu.cn`,

`cwy610@gxust.edu.cn`, `tjia@swu.edu.cn`

## Abstract

Functional brain network (FBN) modeling often relies on local pairwise interactions, whose limitation in capturing high-order dependencies is theoretically analyzed in this paper. Meanwhile, the computational burden and heuristic nature of current hypergraph modeling approaches hinder end-to-end learning of FBN structures directly from data distributions. To address this, we propose to extract high-order FBN structures under global constraints, and implement this as a **G**lobal **C**onstraints oriented **M**ulti-resolution (**GCM**) FBN structure learning framework. It incorporates 4 types of global constraint (signal synchronization, subject identity, expected edge numbers, and data labels) to enable learning FBN structures for 4 distinct levels (sample/subject/group/project) of modeling resolution. Experimental results demonstrate that **GCM** achieves up to a 30.6% improvement in relative accuracy and a 96.3% reduction in computational time across 5 datasets and 2 task settings, compared to 9 baselines and 10 state-of-the-art methods. Extensive experiments validate the contributions of individual components and highlight the interpretability of **GCM**. This work offers a novel perspective on FBN structure learning and provides a foundation for interdisciplinary applications in cognitive neuroscience. Code is publicly available on `https://github.com/lzhan94swu/GCM`.

## 1 Introduction

Functional brain networks (FBNs), which model inter-regional coordination as graphs, have become a central approach for understanding cognition, behavior, and mental disorders [1, 2]. They are typically inferred from multivariate time series (MTS) recorded by fMRI or EEG [3, 4], with the goal of capturing the inter-regional dependencies embedded in neural dynamics [5, 6, 7]. However, because these dependencies are only indirectly observable, reliable network construction remains challenging [8, 9].

Traditional methods construct FBNs by estimating pairwise statistical interactions (e.g., correlation, coherence) between regional signals [10, 11, 12, 13]. Recent machine learning models extract

---

[*]Corresponding author.

39th Conference on Neural Information Processing Systems (NeurIPS 2025).

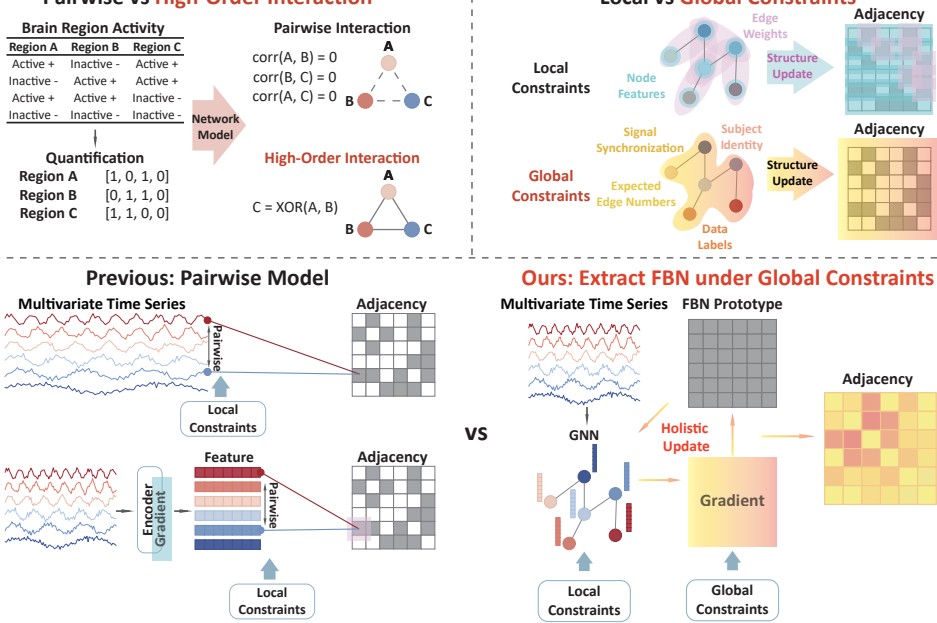

Figure 1: An illustration of the two paradigms for FBN modeling. **Top**: Conceptual diagrams illustrating the core theoretical differences. The top-left provides a concrete example of a high-order (XOR) interaction to demonstrate how pairwise models fail to capture patterns that high-order models can identify. The top-right contrasts the scope of Local Constraints (acting on single nodes or edges) versus Global Constraints (acting on the entire network). **Bottom**: Comparison between the previous pairwise paradigm with our proposed **GCM** framework, which treats the adjacency matrix as a single, learnable object holistically shaped by global objectives.

latent features from MTS via neural encoders, but still rely on pairwise computations to form adjacency matrices for downstream graph analysis [14]. Despite current shift toward end-to-end modeling [8, 15, 16], fully connected graphs could reinforce their performance in prediction according to our empirical study (**Appendix F**). This suggests that the initial structures generated by pairwise methods may be suboptimal or even detrimental. As illustrated by the XOR example in Figure 1 (Top-left), a brain region C might activate only when its two driving regions, A and B, are in opposite states (one active, one inactive)—a relationship that cannot be described by any pairwise correlation between (A,C) or (B,C) alone. This type of irreducible, multi-way dependency is a typical example of a high-order interaction. Recent evidence from hypergraph modeling also suggests that it is necessary to model high-order interactions in the brain [17, 18, 19], which cannot be fully captured by pairwise interactions as proved in **Section 4**. However their inferred FBNs are agnostic to analysis tasks and thus need further statistical processing. Meanwhile, these methods are practically limited by their strong assumptions or prior knowledge demands on dynamics [20, 21, 22] and high computational complexity [23, 24].

Motivated by the observation that high-level cognitive demands shape the organization of functional brain networks [25, 26], we propose to extract FBNs from MTS under global constraints, as illustrated in Figure 1 (Top-right). In this paradigm, we distinguish between two types of constraints. Local constraints operate on a node-pair basis, such as using Pearson correlation to determine a single edge weight, where the final graph is an aggregation of many independent, local decisions. In contrast, global constraints are top-down priors that shape the entire network structure holistically, guiding the optimization of the adjacency matrix as a single entity rather than a collection of independent edges. In this way, FBNs can be learned efficiently and constrained by input MTS and global priors directly, without relying on predefined dynamics [9].

Therefore, we design the **G**lobal **C**onstraints oriented **M**ulti-resolution (**GCM**) framework as an implementation of this paradigm. As illustrated in Figure 1 (Bottom), comparing to previous pairwise models, in **GCM**, FBNs are treated as structured entities optimized under the holistic influence of our global constraints, derived from both data and task semantics, rather than byproducts of local

interactions. To ensure interpretability, **GCM** requires clear semantic specifications on the learned structures and global constraints. As for structure, it is essential to define the modeling resolution, i.e., the unit of representation on which the graph structure operates [27, 28]. Depending on the analytic goal, **GCM** supports four distinct resolutions: sample (e.g., short-term activity [29]), subject (e.g., individual traits [30]), group (e.g., cohort-level patterns [31]), and project (entire dataset, e.g., population-level abstractions [32]). To the best of our knowledge, the semantic distinction between these resolutions has not been well-addressed. However, without such explicit resolution specification the learned FBNs risk semantic ambiguity or misalignment with downstream tasks [33, 34]. Our theoretical and empirical analysis validate the necessity of this formulation.

As a support to the concept of global constraints, a recent work in control theory and cognitive science has proposed a perceptual control architecture [35]. In this framework, organismal functions adapt to environmental disturbances by integrating diverse sensory feedback into coordinated internal responses. Inspired by this view, **GCM** models the brain's functional structure as an adaptive variable shaped by multiple global constraints as signals. These constraints, which we define as global because they influence the entire network structure simultaneously, reflect a distinct form of global feedback, while their joint influence is integrated through a unified gradient. Specifically, our framework integrates four types of global constraints. (1) **Signal Synchronization**: The model is end-to-end, meaning the learned graph is inherently constrained by the high-order temporal dynamics within the raw signals. (2) **Subject Identity**: This constraint imposes a consistent intra-subject signature, or "neural fingerprint," through contrastive regularization [36]. (3) **Expected Edge Numbers**: An adaptive constraint ensures global sparsity in the learned network [37]. (4) **Data Labels**: Task labels guide the structure to align across samples via a supervised loss. These constraint signals are integrated through gradient and fed back through back-propagation, with Gumbel-Sigmoid [38] enabling differentiable sampling over binary graphs. Meanwhile, to retain local co-activation patterns, **GCM** integrates a graph neural network (GNN) as its backbone. Our empirical results report the contribution of each module.

Our contributions are threefold. First, we provide a theoretical proof that pairwise-based models are mathematically incapable of recovering the high-order interactions present in MTS, thereby establishing the necessity of a new paradigm, such as our proposal of learning FBNs under global constraints. Second, we implement this paradigm as **GCM** that directly learns discrete FBNs under global constraints specific to four semantically separated modeling resolutions. Third, extensive empirical evaluation confirms that **GCM** not only improves classification performance but also enhances the interpretability and semantic alignment of the learned FBNs.

## 2 Related Work

Because the idea of high-order FBN structure is addressed in hyper network modeling, and the design of **GCM** is inspired by the ideas of Brain Graph Interpretation (BGI) [39] and Graph Structure Learning (GSL) [40], we briefly introduce the related works along these aspects in this section.

**Hyper Network Modeling**   Research on FBNs aims to model and understand the cooperative functions of the brain and has prospered with the development of complex network theory [6]. Pairwise-based network modeling is a relatively mature field and has been approached from numerous perspectives utilizing diverse tools both in network science and machine learning [9, 41, 42, 43, 44]. Comparatively, hypergraph inference is still in its early stages, but the field is evolving rapidly [22, 45, 46]. Recent efforts span probabilistic modeling grounded in historical connectivity [20, 47] or contagion traces [48], as well as optimization-based formulations adapted to MTS [24]. Despite these advances, their computational overhead presents a major bottleneck for scaling to real-world datasets [23, 24]. More recently, a pioneering work proposes an end-to-end model for learning a fixed number of hyperedges from individual samples [49]. In contrast, our proposed **GCM**framework is explicitly designed to learn a single, generalizable FBN that represents an entire cohort (e.g., at the subject or group resolution), a distinct goal focused on cross-sample interpretability.

**Brain Graph Interpretation**   Brain Graph Interpretation (BGI) methods aim to identify and select important edges in FBNs that contribute most to predictive outcomes [50]. These approaches typically construct FBNs based on pairwise interactions and learn gradient-updated masks to filter structures. Some methods refine the masks dynamically during training [51, 52], others generate

predictive subgraphs to facilitate interpretation [39, 53, 54], integrate multiple modalities for cross-validation [52, 55, 56], or employ attention mechanisms as trainable criteria for edge selection [57, 58, 59]. These methods rely on pairwise-constructed FBNs and apply post-hoc explanations without structural optimization. Consequently, their explanations are often resolution-agnostic, lacking explicit differentiation across semantic levels. In contrast, our framework directly models functional structures under global constraints at multiple modeling resolutions.

**Graph Structure Learning**  Graph Structure Learning (GSL) treats input network structures as noisy or incomplete and aims to refine them based on node features and initial graphs [40]. Traditional GSL research focuses on defending against adversarial attacks [60, 61], later evolving toward task-specific structure optimization [62], mostly for node-level tasks on single graphs [63, 64, 65, 66, 67, 68]. Recent benchmarks [69] highlight that only a few models, such as VIB-GSL [70] and HGP-SL [71], address structure learning across multiple graphs. In brain network modeling, Zong et al. [8] introduced GSL to FBN learning by aligning brain regions using a diffusion model and constructing structures based on feature correlations.

Unlike prior methods, **GCM** supports end-to-end extraction of high-order FBN structures by explicitly encoding modeling resolutions and global constraints, aspects often neglected in existing approaches.

## 3    Preliminaries

We begin by defining a "sample" as the minimal unit of MTS (e.g., a single EEG or fMRI fragment). Formally, each sample is a matrix $\mathbf{X} \in \mathcal{X}$ with shape $N \times T$, where $N$ is the number of nodes and $T$ is the number of time stamps. Each **FBN structure** is an undirected graph $\mathcal{G} = (\mathcal{V}, \mathcal{E})$ with $|\mathcal{V}| = N$, where $\mathcal{V}$ denotes node set and $\mathcal{E}$ denotes edge set.

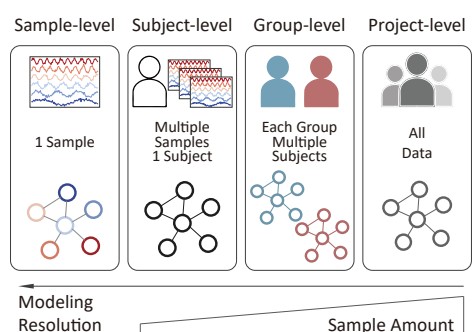

Figure 2: Illustration of the four modeling resolutions defined in this work. Each resolution corresponds to a distinct level of structural abstraction and data aggregation.

**Levels of modeling resolution.**  We consider four levels of modeling resolution, which specify how many samples are aggregated when forming a single FBN:

*Sample level*: Each sample corresponds to one FBN, representing the network within a single recording (e.g. dynamic FBN) [29]. *Subject level*: All samples from the same subject share one FBN, reflecting a relatively stable brain state (e.g. brain fingerprinting) [30]. *Group level*: Samples from a group (e.g., a clinical cohort) form one FBN, capturing a representative pattern of that group (e.g. cognitive states) [31]. *Project level*: The entire dataset yields a single FBN, often used as a statistical template for the project (e.g. brain atlas) [32]. Formally, a research level defines a *mapping function* $\phi : \mathbf{X} \mapsto \mathcal{G}$, which aggregates one or more samples into an FBN. **Appendix B** rigorously proves that sample, subject, group, and project FBNs are semantically non-equivalent.

**FBN Structure Learning.**  We model FBN structure learning problem as extracting connectivity structures from MTS specific to the modeling resolution. Formally, let $\mathcal{P}(\mathcal{X})$ denote the distribution of samples and $\mathcal{P}(\mathcal{Y})$ the corresponding label distribution. Our goal is to learn a parameterized family of graph distributions $\mathcal{P}(\mathcal{G} \mid \phi, \theta(\phi))$, where $\theta(\phi)$ are learnable parameters conditioned on the chosen research level mapping $\phi$. Each distribution is induced by the adjacency matrix distribution $\mathcal{P}(\mathcal{A} \mid \phi, \theta(\phi))$, where adjacency matrices $\mathbf{A} \in \mathbb{R}^{N \times N}$ encode expected connectivity probabilities $\mathbb{E}_{\mathbf{X},y}[p(e_{u,v} \mid \phi, \theta(\phi))]$. Here $e_{u,v}$ denotes an edge between nodes $u$ and $v$ in the FBN, and $p(e_{u,v} \mid \phi, \theta(\phi))$ flexibly models connectivity without assuming specific parametric forms. Thus, by learning $\mathcal{P}(\mathcal{G} \mid \phi, \theta(\phi))$ under different modeling resolutions $\phi$, we enable a *multi-resolution* analysis of functional brain networks.

## 4 The Limitation of Pairwise Network Model

In this section, we formally prove that pairwise network model cannot fully catch the high-order coherence encoded in MTS. Our analysis aligns with and extends prior findings on linear consensus dynamics [72] and triadic interactions [23].

### 4.1 Definitions

Let $\mathbf{X}(t) = \begin{bmatrix} X_1(t), \dots, X_N(t) \end{bmatrix}^\top$ be a $N$–dimensional, zero-mean, finite-variance MTS. For each lag $\omega \in \mathbb{Z}$ denote the second-order moment $\Sigma_{ij}(\omega) = \mathrm{Cov}\big[X_i(t), X_j(t+\omega)\big]$. Higher-order joint structure is captured by the $k$-th order *cumulant*[2]

$$\kappa_{i_1\dots i_k}(\omega_1, \dots, \omega_{k-1}) = \mathrm{cum}\big[X_{i_1}(t), X_{i_2}(t+\omega_1), \dots, X_{i_k}(t+\omega_{k-1})\big]. \tag{1}$$

**Proof.** A pairwise network is an adjacency matrix $\mathbf{A} \in \mathbb{R}^{N \times N}$ whose entries are generated by $A_{ij} = f_{ij}(X_i, X_j)$, where $f_{ij}$ is a specific edge rule depending only on the bivariate distribution of $(X_i, X_j)$.

**Higher-order dependence.** We say that $\mathbf{X}$ exhibits *higher-order dependence* if there exists $k \geq 3$ and indices $i_1, \dots, i_k$ such that $\kappa_{i_1\dots i_k}(\cdot) \neq 0$.

### 4.2 The Expressive Limit of Pairwise Network Models

**Lemma 1** (Second-order sufficiency). *A random vector $\mathbf{Z}$ is multivariate Gaussian* iff *all cumulants of order $k \geq 3$ vanish* [73]. *Consequently, only for Gaussian data is the joint distribution fully determined by second-order statistics.*

**Theorem 1** (The limitation of Pairwise Network Model). *Let $\mathbf{X}^{(1)}$ and $\mathbf{X}^{(2)}$ be two d-dimensional time series $(d \geq 3)$ satisfying $\Sigma_{ij}^{(1)}\omega) = \Sigma_{ij}^{(2)}(\omega), \forall i, j, \forall \omega \in \Omega,$ for some finite lag set $\mathcal{T}$ actually used by the edge rule. Assume further that there exists $k \geq 3$ with*

$$\kappa_{i_1\dots i_k}^{(1)}(\cdot) \neq \kappa_{i_1\dots i_k}^{(2)}(\cdot) \quad \text{for some } i_1, \dots, i_k. \tag{2}$$

*Then **every** pairwise network generator $A_{ij} = f_{ij}(X_i, X_j)$ produces $\mathbf{A}^{(1)} = \mathbf{A}^{(2)}$, while the two joint processes $\mathbf{X}^{(1)} \overset{\mathrm{law}}{\neq} \mathbf{X}^{(2)}$. (Proof in **Appendix C**).*

**Corollary 1.** **Theorem** 1 *remains valid for* weighted *graphs (let g be identity), directed graphs, or any edge rule that aggregates over a finite lag set $\Omega$.*

## 5 Methodology

We introduce the proposed **GCM** in this section. It consists of three modules: (i) a prototype-based graph generator with Gumbel–Sigmoid relaxation, (ii) a batch-wise binarization algorithm (**BBA**) that enforces hard edge sparsity, and (iii) a multi-objective training loss aligned with task labels, subject identity, and sparsity priors. **Appendix E** summarizes pseudocode for **GCM** and **BBA**.

**Prototype Graph and Continuous Relaxation.** We learn a binarized functional brain network (FBN) $\mathbf{A} \in \{0, 1\}^{N \times N}$ from the conditional distribution $\mathcal{P}(\mathcal{A} \mid \phi, \theta(\phi))$, where $\phi$ parameterises the graph sampler and $\theta(\phi)$ denotes GNN weights. Instead of drawing $\mathbf{A}$ directly, **GCM** first samples a prototype matrix $\hat{\mathbf{A}} \sim \mathcal{P}(\hat{\mathcal{A}} \mid \phi, \theta(\phi))$ and symmetrises it by $\mathbf{A} = \sigma\big(\hat{\mathbf{A}} + \hat{\mathbf{A}}^\top\big)$, where $\sigma(\cdot)$ is the element-wise sigmoid so that $\mathbf{A}_{ij} \in [0, 1]$. We initialise $\hat{\mathbf{A}}_{ij} \overset{\mathrm{i.i.d.}}{\sim} \mathcal{U}(0, 1)$.

To enable back-propagation through discrete edges, we adopt the Gumbel–Softmax (Concrete) relaxation [38]. Let $\mathbf{M}_{ij} = -\log(-\log u)$, $u \sim \mathcal{U}(0, 1)$; the relaxed adjacency is $\tilde{\mathbf{A}} = \sigma((\mathbf{A} + \mathbf{M})/\tau)$ where $\tau$ is the temperature. Gradients w.r.t. $\hat{\mathbf{A}}$ and $\theta$ are obtained via the chain rule.

---

[2]Any other faithful measure of $k$-variable dependence (e.g. joint mutual information, total correlation) would serve equally well.

**Batch Binarization Algorithm (BBA).** Given a mini-batch $\tilde{\mathbf{A}} \in \mathbb{R}^{B \times N \times N}$, **BBA** keeps the $k^{(b)}$ largest (row-major flatted) entries in each $\tilde{\mathbf{A}}^{(b)}$, where $k^{(b)} = \frac{1}{2} \sum_{i=1}^{N} \sum_{j \neq i} \tilde{\mathbf{A}}^{(b)}$ is the expected edge number of $\tilde{\mathbf{A}}^{(b)}$ updated during the training process to ensure an adaptive constraint of sparsity, and $b$ is the index of network within the batch:

$$\mathring{\mathbf{Q}}^{(b)} = \mathbf{m}^{(b)} \odot \mathbf{Q}^{(b)}, \quad \mathbf{m}_j^{(b)} = \mathbf{1}\{j \in \text{Top-}k^{(b)}\}, \tag{3}$$

followed by reshaping back to obtain a binarized batch $\{\mathcal{G}^{(b)}\}_{b=1}^{B}$.

**Graph Representation Learning.** For each sample $(\mathbf{X}, \mathcal{G})$, node signals $\mathbf{X} \in \mathbb{R}^{N \times T}$ are fed to a $L$-layer GNN backbone [74]. After message passing, a permutation-invariant readout $\rho(\cdot)$ (mean, sum, max, or Transformer pooling [75]) produces a graph embedding $\mathbf{e} \in \mathbb{R}^{d_e}$.

**Training Objectives.** *Label-aligned loss:* A linear classifier predicts class logits $\hat{\mathbf{y}}_i = \text{softmax}(\mathbf{W}^\top \mathbf{e}_i + \mathbf{b})$, optimised by cross-entropy $\mathcal{L}_1 = -\sum_i y_i \log \hat{y}_i$. *Subject identity contrast:* Samples from the same subject are positive, others negative:

$$\mathcal{L}_2 = -\frac{1}{|\mathcal{P}_{\text{pos}}|} \sum_{(i,j) \in \mathcal{P}_{\text{pos}}} \log \sigma\left(\frac{\mathbf{e}_i^\top \mathbf{e}_j}{\tau_{cl}}\right) - \frac{1}{|\mathcal{P}_{\text{neg}}|} \sum_{(i,k) \in \mathcal{P}_{\text{neg}}} \log \sigma\left(-\frac{\mathbf{e}_i^\top \mathbf{e}_k}{\tau_{cl}}\right), \tag{4}$$

with temperature $\tau_{cl}$. *Sparse prior:* We impose an $\ell_1$ surrogate on $\hat{\mathbf{A}}$ as $\mathcal{R} = \left\|\hat{\mathbf{A}}\right\|_1$. The total objective is $\mathcal{L} = \mathcal{L}_1 + \alpha\mathcal{L}_2 + \beta\mathcal{R}$, optimised jointly over $(\theta, \hat{\mathbf{A}})$ using Adam.

**Theorem 2** (Existence of a Higher-Order-Exact **GCM**). *Let $\mathbf{X}(t) \in \mathbb{R}^N$ be a stochastic process and $y \in \{1, \ldots, C\}$ obey*

$$y = g_\star\big(\psi\big(X_{i_1}(t), \ldots, X_{i_k}(t)\big)\big), \quad k \geq 3, \tag{5}$$

*where (i) $\psi : \mathbb{R}^k \to \mathbb{R}$ is* measurable*; (ii) $g_\star$ is a* class-separable *measurable map (assuming $\exists$ disjoint measurable pre-images for each class). For any $\varepsilon > 0$ there exist $\phi^\varepsilon, \theta^\varepsilon$, temperature $\tau^\varepsilon > 0$, and depth $L = \max\{1, \text{diam}(\mathbf{A}^\star)\}$ such that the **GCM** classifier $f_{\theta^\varepsilon, \phi^\varepsilon}$ satisfies*

$$\Pr\big[f_{\theta^\varepsilon, \phi^\varepsilon}(\mathbf{X}(t)) \neq y\big] \leq \varepsilon.$$

This theory ensures the existence and convergence of **GCM**. Our formal proof of **Theorem** 2 in **Appendix D** shows that **GCM** can recover arbitrary high-order rules. These justify both our multi-resolution design and the use of global optimization.

## 6 Experiments

To demonstrate the advantage of **GCM**, we utilize it with other baselines in each classification task. The FBN structure learned by **GCM** is used as the computing graph for a GNN classifier. If **GCM** outperforms other baselines using the same GNN layer, the difference can only be explained by the underlying network structure, hence proving that **GCM** learns a more accurate and comprehensive representation of FBN. For the task at the sample level, we also consider several none-GNN based methods for their prevalence in neuroscience. The performance advances of **GCM** over them validate the benefits of combining GNN with extracting high-order network information.

Table 1: Statistics of the Data Sets

|  | #sample | #subject | #class | #node | #feature | Class-Type |
|---|---|---|---|---|---|---|
| DynHCP$_{\text{Activity}}$ | 260,505 | 7,443 | 7 | 100 | 100 | sample |
| DynHCP$_{\text{Age}}$ | 36,278 | 1,067 | 3 | 100 | 100 | subject |
| DynHCP$_{\text{Gender}}$ | 36,720 | 1,080 | 2 | 100 | 100 | subject |
| Cog State | 90,000 | 60 | 5 | 61 | 300 | sample |
| SLIM | 3,246 | 541 | 2 | 236 | 30 | subject |

### 6.1 Experiment Settings

**Data** We use 5 open-source real-world datasets, including DynHCP Activity/Age/Gender [76], Cog State [77], and SLIM [78]. These datasets cover both fMRI and EEG, the most commonly used brain imaging modalities, for testing broad applicability. The overall statistics of the data are summarized in Table 1. Description and preprocessing of the data are detailed in **Appendix J**.

**Baselines** We choose traditional methods (**LR**, **SVM**, **Random Forest** (**RF**), **XGBoost**, **MLP**), which are commonly used in neuroscience domain, to establish baseline performance. GNN families (**GCN** [74], **SAGE** [79], **GAT** [80], **GIN** [81]) are included to highlight the improvement of **GCM** against GNN backbones. To position our work within the current frontier of structured graph learning, we include representative SOTA methods from both the BGI line (**IBGNN** [82], **IBGNN+** [82], **IGS** [51], **IC-GNN** [53], **BrainIB** [39]) and the GSL line (**VIB-GSL** [70], **HGP-SL** [71], **DGCL** [8]). To further benchmark our method against the SOTAs, we incorporate two key baselines. First, we include the Brain Network Transformer (**BNT**) [83] due to its remarkable predictive performance. Second, we include **Hybrid** [49], a conceptually related model that learns a fixed quantity of high-order interactions. (Details in **Appendix K**)

**Tasks** Brain activity prediction tasks are commonly categorized into inter-subject and intra-subject classification [84, 85]. In inter-subject classification, all subjects and their samples are split into training, validation, and test sets, with predictions targeting the test set subjects' labels. In intra-subject classification, each subject's samples are split into training, validation, and test sets, with predictions focused on the test set labels. We align the baselines with the proposed four modeling resolution levels for fair comparison, as defined in **Preliminaries**, based on their design. For all levels, the task is to predict the label of each individual sample. For GNN-based methods, the learned FBNs used as computation graphs, where each sample serves as the feature of a node. **Appendix J** Contains more detailed reproducibility settings such as hyper parameters and experiment environment.

## 6.2 Experimental Results

We report the most important results to support the superior performance of the proposed **GCM** and the interpretability of the learned FBN in the main body. We also provide more detailed analyses including **The Role of Initial Structure**(**Appendix F**), **Results Using Additional Metrics** (**Appendix G**), **FBN Structure Visualization**(**Appendix I**) and **Sensitivity Analysis**(**Appendix H**) for a more comprehensive assessment of **GCM**.

Table 2: Sample-Level Structure Learning Results (Accuracy$_{\pm\text{Std}}$)

| Method | DynHCP$_{\text{Activity}}$ | | DynHCP$_{\text{Age}}$ | | DynHCP$_{\text{Gender}}$ | | CogState | | SLIM | |
|---|---|---|---|---|---|---|---|---|---|---|
| | Intra | Inter | Intra | Inter | Intra | Inter | Intra | Inter | Intra | Inter |
| LR | $0.939_{\pm0.026}$ | $0.821_{\pm0.091}$ | $0.809_{\pm0.008}$ | $0.357_{\pm0.061}$ | $0.879_{\pm0.005}$ | $0.625_{\pm0.092}$ | $0.276_{\pm0.025}$ | $0.218_{\pm0.066}$ | $0.521_{\pm0.013}$ | $0.539_{\pm0.070}$ |
| RF | $0.945_{\pm0.002}$ | $0.825_{\pm0.001}$ | $0.812_{\pm0.001}$ | $0.409_{\pm0.002}$ | $0.824_{\pm0.001}$ | $0.629_{\pm0.001}$ | $0.288_{\pm0.001}$ | $0.264_{\pm0.001}$ | $0.635_{\pm0.003}$ | $0.626_{\pm0.003}$ |
| XGBoost | $0.959_{\pm0.001}$ | $0.832_{\pm0.001}$ | $0.743_{\pm0.002}$ | $0.388_{\pm0.001}$ | $0.798_{\pm0.001}$ | $0.611_{\pm0.002}$ | $0.307_{\pm0.003}$ | $0.282_{\pm0.002}$ | $0.623_{\pm0.002}$ | $0.610_{\pm0.002}$ |
| SVM | $0.952_{\pm0.035}$ | $0.842_{\pm0.008}$ | $0.827_{\pm0.017}$ | $0.394_{\pm0.016}$ | $0.882_{\pm0.011}$ | $0.631_{\pm0.072}$ | $0.321_{\pm0.018}$ | $0.270_{\pm0.041}$ | $0.567_{\pm0.025}$ | $0.554_{\pm0.014}$ |
| MLP | $0.951_{\pm0.014}$ | $\underline{0.847}_{\pm0.010}$ | $0.788_{\pm0.019}$ | $0.372_{\pm0.032}$ | $0.903_{\pm0.009}$ | $0.643_{\pm0.074}$ | $\underline{0.376}_{\pm0.009}$ | $0.273_{\pm0.076}$ | $0.518_{\pm0.022}$ | $0.549_{\pm0.042}$ |
| GCN | $0.774_{\pm0.022}$ | $0.736_{\pm0.068}$ | $0.423_{\pm0.022}$ | $\underline{0.455}_{\pm0.053}$ | $0.650_{\pm0.017}$ | $0.543_{\pm0.045}$ | $0.290_{\pm0.023}$ | $0.303_{\pm0.014}$ | $0.552_{\pm0.012}$ | $0.543_{\pm0.072}$ |
| SAGE | $0.791_{\pm0.005}$ | $0.794_{\pm0.032}$ | $0.464_{\pm0.019}$ | $0.411_{\pm0.034}$ | $0.633_{\pm0.025}$ | $0.594_{\pm0.060}$ | $0.334_{\pm0.012}$ | $0.301_{\pm0.022}$ | $\underline{0.657}_{\pm0.013}$ | $0.596_{\pm0.053}$ |
| GAT | $0.140_{\pm0.000}$ | $0.138_{\pm0.000}$ | $0.212_{\pm0.000}$ | $0.202_{\pm0.004}$ | $0.542_{\pm0.001}$ | $0.532_{\pm0.004}$ | $0.296_{\pm0.010}$ | $0.305_{\pm0.009}$ | $0.622_{\pm0.029}$ | $0.602_{\pm0.018}$ |
| GIN | $0.480_{\pm0.062}$ | $0.495_{\pm0.058}$ | $0.449_{\pm0.013}$ | $0.401_{\pm0.065}$ | $0.572_{\pm0.012}$ | $0.565_{\pm0.030}$ | $0.308_{\pm0.017}$ | $0.280_{\pm0.023}$ | $0.534_{\pm0.023}$ | $0.465_{\pm0.040}$ |
| VIB-GSL | $0.925_{\pm0.006}$ | $0.772_{\pm0.017}$ | $0.600_{\pm0.060}$ | $0.429_{\pm0.031}$ | $0.781_{\pm0.029}$ | $0.574_{\pm0.034}$ | $0.313_{\pm0.014}$ | $0.303_{\pm0.010}$ | $0.529_{\pm0.007}$ | $0.537_{\pm0.013}$ |
| HGP-SL | $0.715_{\pm0.010}$ | $0.647_{\pm0.053}$ | $0.529_{\pm0.012}$ | $0.404_{\pm0.022}$ | $0.766_{\pm0.006}$ | $0.612_{\pm0.003}$ | $0.286_{\pm0.005}$ | $0.270_{\pm0.008}$ | $0.556_{\pm0.013}$ | $0.552_{\pm0.009}$ |
| DGCL | $0.831_{\pm0.013}$ | $0.631_{\pm0.017}$ | $0.758_{\pm0.037}$ | $0.392_{\pm0.033}$ | $0.860_{\pm0.001}$ | $0.615_{\pm0.017}$ | $0.316_{\pm0.011}$ | $0.275_{\pm0.007}$ | $0.552_{\pm0.009}$ | $0.539_{\pm0.017}$ |
| IBGNN | $0.908_{\pm0.004}$ | $0.586_{\pm0.278}$ | $0.768_{\pm0.025}$ | $0.397_{\pm0.008}$ | $0.858_{\pm0.027}$ | $0.591_{\pm0.025}$ | $0.343_{\pm0.008}$ | $0.300_{\pm0.012}$ | $0.648_{\pm0.031}$ | $0.617_{\pm0.027}$ |
| IC-GNN | $0.878_{\pm0.053}$ | $0.744_{\pm0.092}$ | $0.678_{\pm0.105}$ | $0.425_{\pm0.024}$ | $0.851_{\pm0.065}$ | $0.658_{\pm0.067}$ | $0.318_{\pm0.077}$ | $0.313_{\pm0.053}$ | $0.638_{\pm0.103}$ | $0.614_{\pm0.072}$ |
| BrainIB | $0.926_{\pm0.043}$ | $0.839_{\pm0.018}$ | $0.818_{\pm0.035}$ | $0.445_{\pm0.019}$ | $0.873_{\pm0.011}$ | $0.662_{\pm0.020}$ | $0.332_{\pm0.013}$ | $\underline{0.325}_{\pm0.021}$ | $0.644_{\pm0.023}$ | $\underline{0.631}_{\pm0.012}$ |
| BNT | $\mathbf{0.974}_{\pm0.002}$ | $0.839_{\pm0.002}$ | $\mathbf{0.945}_{\pm0.004}$ | $0.386_{\pm0.004}$ | $\mathbf{0.976}_{\pm0.003}$ | $\mathbf{0.678}_{\pm0.008}$ | $\mathbf{0.396}_{\pm0.006}$ | $0.337_{\pm0.015}$ | $0.553_{\pm0.012}$ | $0.529_{\pm0.007}$ |
| Hybrid | $0.801_{\pm0.013}$ | $0.766_{\pm0.023}$ | $0.525_{\pm0.008}$ | $0.442_{\pm0.015}$ | $0.788_{\pm0.007}$ | $0.592_{\pm0.009}$ | $0.293_{\pm0.017}$ | $0.286_{\pm0.021}$ | $0.537_{\pm0.008}$ | $0.522_{\pm0.018}$ |
| **GCM**$_{\text{sample}}$ | $\underline{0.963}_{\pm0.001}$ | $\mathbf{0.852}_{\pm0.021}$ | $\underline{0.853}_{\pm0.005}$ | $\mathbf{0.463}_{\pm0.017}$ | $\underline{0.937}_{\pm0.002}$ | $\underline{0.668}_{\pm0.022}$ | $0.352_{\pm0.014}$ | $\mathbf{0.356}_{\pm0.021}$ | $\mathbf{0.680}_{\pm0.014}$ | $\mathbf{0.658}_{\pm0.035}$ |

**Multi-Resolution Structure Learning: Sample Level** The sample level is the most common scenario in brain activity prediction, as it focuses on individual samples and their corresponding brain network structures. At the sample level, **GCM** consistently outperforms or matches traditional methods and SOTAs as shown in Table 2. These results highlight that **GCM** excels in capturing the underlying brain activity structure, surpassing Pearson-based FBNs, which are limited by their reliance on linear correlations. On the one hand, the results prove that catching higher-order synchronization of MTS is beneficial to downstream prediction. On the other hand, this demonstrates the effectiveness of **GCM** in learning a more robust and flexible structure that better represents the data's complexity. GSL and BGI SOTAs gain an overall superiority against GNN-based methods, providing the necessity of FBN structure learning. An interesting phenomenon is that traditional methods exhibit a strong competitiveness on several datasets. This explains why these methods are still popular in current neuroscience domain.

Table 3: Multi-Resolution Structure Learning Results (Accuracy$_{\pm\text{Std}}$)

| Method | DynHCP$_{\text{Activity}}$ | | DynHCP$_{\text{Age}}$ | | DynHCP$_{\text{Gender}}$ | | CogState | | SLIM | |
|---|---|---|---|---|---|---|---|---|---|---|
| | Intra | Inter | Intra | Inter | Intra | Inter | Intra | Inter | Intra | Inter |
| IGS | $0.861_{\pm0.005}$ | $0.856_{\pm0.004}$ | $0.808_{\pm0.022}$ | $0.460_{\pm0.034}$ | $0.882_{\pm0.138}$ | $0.608_{\pm0.047}$ | $0.326_{\pm0.005}$ | $0.323_{\pm0.011}$ | $0.541_{\pm0.013}$ | $0.652_{\pm0.027}$ |
| GCM$_{\text{subject}}$ | $\mathbf{0.963}_{\pm0.002}$ | $\mathbf{0.857}_{\pm0.019}$ | $\mathbf{0.889}_{\pm0.005}$ | $\mathbf{0.473}_{\pm0.031}$ | $\mathbf{0.939}_{\pm0.002}$ | $\mathbf{0.670}_{\pm0.019}$ | $\mathbf{0.413}_{\pm0.008}$ | $\mathbf{0.364}_{\pm0.007}$ | $\mathbf{0.681}_{\pm0.005}$ | $\mathbf{0.674}_{\pm0.010}$ |
| IBGNN+ | $0.765_{\pm0.301}$ | $0.453_{\pm0.346}$ | $0.786_{\pm0.033}$ | $0.380_{\pm0.020}$ | $0.881_{\pm0.005}$ | $0.596_{\pm0.024}$ | $0.356_{\pm0.012}$ | $0.315_{\pm0.014}$ | $\mathbf{0.667}_{\pm0.023}$ | $\mathbf{0.644}_{\pm0.040}$ |
| GCM$_{\text{project}}$ | $\mathbf{0.969}_{\pm0.002}$ | $\mathbf{0.869}_{\pm0.004}$ | $\mathbf{0.878}_{\pm0.002}$ | $\mathbf{0.462}_{\pm0.029}$ | $\mathbf{0.933}_{\pm0.007}$ | $\mathbf{0.674}_{\pm0.028}$ | $\mathbf{0.437}_{\pm0.010}$ | $\mathbf{0.419}_{\pm0.023}$ | $0.655_{\pm0.011}$ | $0.644_{\pm0.034}$ |
| GCM$_{\text{group}}$ | $0.971_{\pm0.001}$ | $0.855_{\pm0.007}$ | $0.891_{\pm0.007}$ | $0.489_{\pm0.029}$ | $0.958_{\pm0.002}$ | $0.686_{\pm0.025}$ | $0.493_{\pm0.008}$ | $0.454_{\pm0.013}$ | $0.692_{\pm0.046}$ | $0.694_{\pm0.034}$ |

**Multi-Resolution Structure Learning: Subject, Group, and Project Levels** Comparing to sample level analysis, methods in these three levels are less discussed in previous FBN structure learning works. Thus only two competitors respectively aiming for subject-level and project-level are included. As shown in Table 3, **GCM** outperforms these two methods on most conditions. This not only shows that **GCM** adapts seamlessly to higher-level modeling resolutions, but also proves that encoding and utilizing global constraints is crucial for tasks in these level of resolutions. It should be noted that we isolate the results of group-level **GCM**. On the one hand, there is no previous discussion particularly aiming at this scenario. On the other hand, FBN learned in this scenario may suffer the argument of label leakage under the context of machine learning settings. However, the target in this scenario is to learn possible FBN structures that can better reflect the structural distinction between categories. Thus, the accuracy is not only a judgement for method, but an index to investigate the confidence that the learned FBNs reflect the non-linear relationships between the input data and labels. From this point of view, although **GCM**$_{\text{group}}$ literally outperforms all the baselines, there's a large room for improvement to learn reliable FBNs in this scenario.

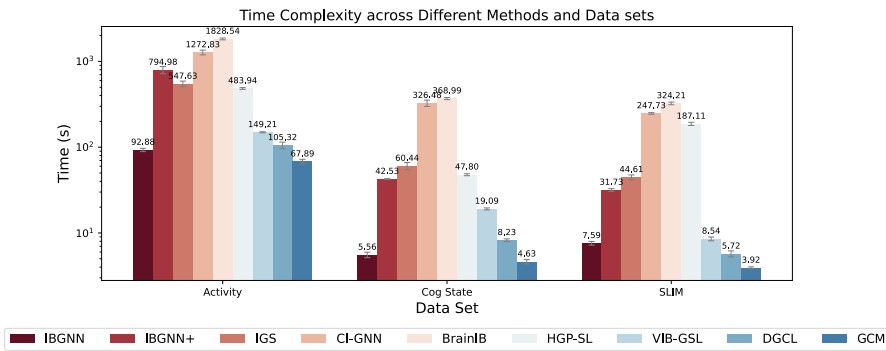

Figure 3: Time consumption of each method.

**Computational Analysis** **GCM**'s computational efficiency is primarily determined by the GNN component (here, **SAGE**), incurring $\mathcal{O}(N^2)$ complexity. Figure 3 compares computation times of BGI and GSL methods, including **GCM**, across datasets. All **GCM** variants share identical costs, so they appear as one category. We omit DynHCP$_{\text{Age}}$ and DynHCP$_{\text{Gender}}$ since they match DynHCP$_{\text{Activity}}$ in graph dimensions and SLIM in class counts. The superior efficiency of **GCM** is clearly reflected in the results, owing to its streamlined design.

### 6.3 Ablation Studies

**Impact of Batch Binarization Algorithm (BBA)** Table 4 presents an ablation study of our **BBA** to validate its key design choices. First, the binarized **GCM** model consistently outperforms a non-binarized "Dense" variant across most tasks and resolutions. This confirms that binarization acts as a crucial regularizer, forcing the model to learn a sparse structural scaffold and preventing overfitting. Second, **GCM** with its adaptive **BBA** also surpasses all fixed-threshold methods. While the performance of fixed thresholds plateaus as density increases, **BBA** avoids this issue by adaptively tuning sparsity for each FBN; nonetheless, a $5\%$ fixed density offers a reasonable trade-off for simpler applications. These advantages are consistently observed across all resolutions (sample, subject, group, and project), validating the **BBA** module as a critical and robust component for learning high-performing FBN structures.

Table 4: Ablation Study of Structural Modeling Choices (ACC)

| Dataset | Task | Sample-level Ablations | | | | | | | Subject-level | | Group-level | | Project-level | |
|---|---|---|---|---|---|---|---|---|---|---|---|---|---|---|
| | | Dense | 0.05% | 0.1% | 0.5% | 1% | 5% | **GCM** | Dense | **GCM** | Dense | **GCM** | Dense | **GCM** |
| DynHCP$_{Activity}$ | Inter | 0.843 | 0.792 | 0.808 | 0.827 | 0.835 | 0.837 | **0.852** | 0.849 | **0.857** | **0.886** | 0.855 | 0.857 | **0.869** |
| | Intra | 0.950 | 0.895 | 0.927 | 0.943 | 0.949 | 0.951 | **0.963** | 0.962 | **0.963** | 0.971 | 0.971 | 0.959 | **0.969** |
| DynHCP$_{Age}$ | Inter | 0.408 | 0.381 | 0.402 | 0.418 | 0.420 | 0.428 | **0.463** | 0.425 | **0.473** | 0.432 | **0.489** | 0.418 | **0.462** |
| | Intra | 0.790 | 0.828 | 0.838 | 0.837 | 0.842 | 0.845 | **0.853** | 0.848 | **0.889** | 0.891 | **0.896** | 0.794 | **0.878** |
| DynHCP$_{Gender}$ | Inter | 0.657 | 0.605 | 0.628 | 0.631 | 0.633 | 0.653 | **0.668** | 0.662 | **0.670** | 0.673 | **0.686** | 0.637 | **0.674** |
| | Intra | 0.889 | 0.883 | 0.891 | 0.912 | 0.925 | 0.928 | **0.937** | 0.921 | **0.939** | **0.961** | 0.958 | 0.894 | **0.933** |
| Cog State | Inter | **0.359** | 0.311 | 0.313 | 0.321 | 0.325 | 0.327 | 0.356 | 0.357 | **0.364** | 0.345 | **0.493** | 0.350 | **0.419** |
| | Intra | **0.356** | 0.322 | 0.333 | 0.343 | 0.348 | 0.348 | 0.352 | 0.392 | **0.413** | 0.482 | **0.493** | 0.399 | **0.437** |
| SLIM | Inter | 0.656 | 0.604 | 0.622 | 0.631 | 0.632 | 0.639 | **0.658** | 0.670 | **0.674** | 0.683 | **0.694** | **0.657** | 0.644 |
| | Intra | 0.661 | 0.618 | 0.632 | 0.637 | 0.645 | 0.649 | **0.680** | 0.665 | **0.681** | 0.677 | **0.692** | **0.670** | 0.655 |

**Impact of Subject Consistency Contrast**   The impact of subject identity is reflected by training strategies on **GCM**. Results are shown in Figure 4. Specifically, **GCMi** represents **GCM** without individual consistency loss $\mathcal{L}_2$, and **GCMd** refers to **GCM** without sparsity regularization $\mathcal{R}$. Removing $\mathcal{L}_2$ and $\mathcal{R}$ results in a moderate performance drop under most conditions. However, for some settings, such as in group level and project level prediction in inter-subject prediction, the degradation becomes unbearable. This emphasizes the necessity of maintaining subject consistency and sparsity in the FBN structure.

## 6.4   Case Study: Sex Difference

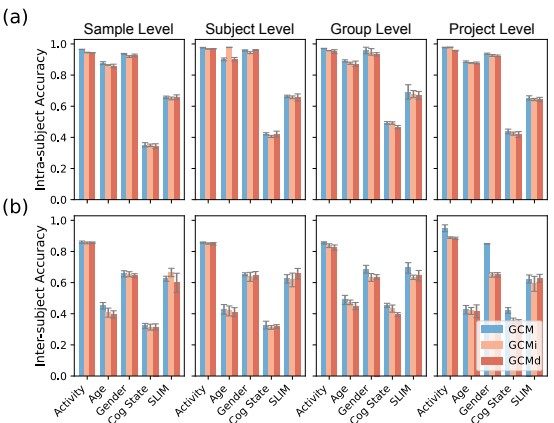

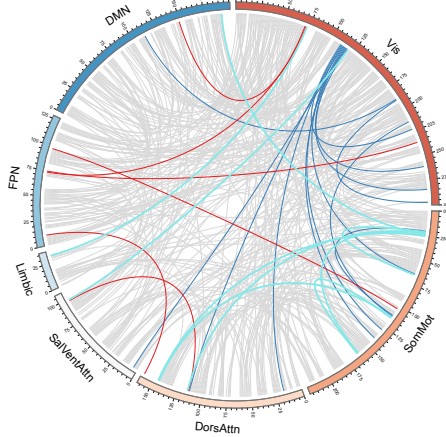

Figure 4: Accuracy of **GCM** using different training strategies.

Figure 5: FBN of DynHCP$_{Gender}$ combining four modeling resolutions, with brain regions as chunks and edges as chords.

To illustrate the necessity and interpretability of our proposed four-level brain network division, we visualize FBNs learned by **GCM** on DynHCP$_{Gender}$ in the intra-subject setting, where model confidence (accuracy) is highest. For sample and subject levels, we average the networks of male and female subjects to obtain group-wise representations. Following the original release [76], we retain the top 10% of edges by weight and visualize combined networks at the sample, subject, and group levels. ROIs are grouped into seven brain regions based on the Schaefer atlas [86].

As shown in Figure 5, the common edges between male and female specific to each resolution are depicted in gray. The edges particularly shown in female and male across resolutions are depicted in red and blue, respectively. Common edges between female and male across resolutions are highlighted in cyan. The results align with previous studies in the following three aspects: (1) Higher-order synergies exhibit non-random organization, integrating multiple brain regions into coherent systems that appear independent under conventional correlation-based analyses [17]. (2) Under each modeling resolution, most of the edges are shared among female and male [87]. (3) Male FBN shows higher edge weights in edges linking to vision networks [88] while female FBN shows a more complex

pattern [89]. The limited amount of common edges sharing across different levels further shows the uninterchangeability between modeling resolutions as illustrated in **Theorem 3**, which is the first time to be clarified to the best of our knowledge. Comparisons between FBNs learned by **GCM** and baselines are provided in **Appendix I**.

# 7   Convergence Analysis

To demonstrate the convergence of **GCM** on intra- and inter-subject classification tasks, as proved in **Theorem** 2, we present the training loss curves for cross-entropy loss ($\mathcal{L}_1$) and contrastive loss ($\mathcal{L}_2$) in Figure 6. Both losses eventually converge on each dataset. It is interesting that $\mathcal{L}_1$ shows temporary vibrations, aligning with the structural reconfiguration and path reselection process observed in FBN evolution [35]. $\mathcal{L}_2$ exhibits a slower, linear decline, reflecting a cold start in subject-level learning. These results highlight opportunities in future to enhance the utilization of global constraint.

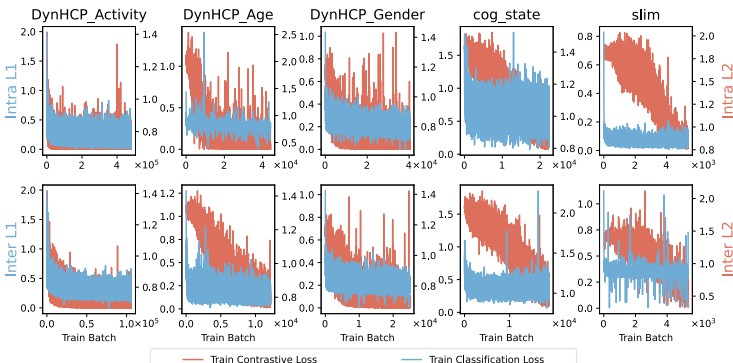

Figure 6: Training loss of **GCM** for each dataset reaches a steady state after training on several batches.

# 8   Discussion and Conclusion

Our results support the theoretical expectation that global constraints enhance structural guidance, surpassing pairwise-based methods in aligning information flow across raw data, macro attributes, and the learned FBNs. The learned high-order patterns align with previous conclusions in hyper graph modeling, proves the interpretability of the learned FBNs. Notably, we observe that functional brain networks learned at different resolutions exhibit distinct structural patterns, validating that these modeling resolutions are not interchangeable. While the proposed **GCM** framework offers strong empirical and theoretical benefits, it also has limitations and potential societal impacts as summarized in **Appendix L** and **Appendix M**, respectively.

Instead of treating prediction accuracy as a unified metric for model performance [82], we reinterpret accuracy as a confidence index that reflects the reliability of the learned FBNs. This formulation offers a principled way to compare models across different analytical objectives and renders overfitted FBNs at the group level as deliberate and meaningful modeling outcomes rather than incorrect label leakage. This further extends traditional statistical comparisons of individual- and population-level connectivity into a unified structural learning framework [90, 91].

Moreover, our study offers a theoretically grounded implementation of the proposed global updating modeling, which provides a reliable framework for FBN structure learning. By analyzing the limitations of existing approaches, we provide a theoretical complement to MTS-based network modeling field [3, 92]. Moreover, by exploiting how evaluation metrics are utilized, and by rethinking overfitting phenomena in group-level modeling as a reliable modeling results, our work introduces a novel lens for applying machine learning in neuroscience and cognitive science studies. Future work includes addressing the limiations above, or extending **GCM** to multi-modal and multi-view scenarios [93]. Further exploration on this topic not only facilitates brain science but also sheds lights on interdisciplinary research.

## Acknowledgements

This work is supported by Natural Science Foundation of China (No. 62402398, No. 72374173), the Fundamental Research Funds for the Central Universities (No. SWU-XDJH202303, No. SWU-KR24025), University Innovation Research Group of Chongqing (No. CXQT21005), China Scholarship Council (CSC) program (No. 202306990091, No. 202406990056) and Chongqing Graduate Research Innovation Project (No. CYB22129, No. CYB240088). The experiments are supported by the High Performance Computing clusters at Southwest University.

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

# A  Notation Summary

Table A1: Summary of notations used throughout the paper.

| Symbol | Description |
|---|---|
| *General & Graph Theory* | |
| $N$ | Number of nodes (brain regions) in a graph. |
| $T$ | Number of time stamps in a single MTS sample. |
| $\mathcal{G} = (\mathcal{V}, \mathcal{E})$ | An undirected graph representing an FBN structure. |
| $\mathcal{V}, \mathcal{E}$ | The set of nodes and edges in a graph, respectively. |
| $e_{u,v}$ | An edge between nodes $u$ and $v$. |
| *Input Data & Modeling Resolutions* | |
| $\mathbf{X} \in \mathbb{R}^{N \times T}$ | A multivariate time series (MTS) sample. |
| $\mathbf{X}(t)$ | An $N$-dimensional MTS at time $t$. |
| $\phi(\cdot)$ | A mapping function that defines the modeling resolution. |
| $\mathcal{P}(\mathcal{X}), \mathcal{P}(\mathcal{Y})$ | The distributions of samples and their corresponding labels. |
| *GCM Framework Components* | |
| $\mathbf{A} \in \{0,1\}^{N \times N}$ | The binary adjacency matrix of a learned FBN. |
| $\hat{\mathbf{A}}$ | The learnable, continuous prototype matrix for the graph generator. |
| $\mathbf{M}$ | A matrix of Gumbel noise used for Gumbel-Softmax relaxation. |
| $\tilde{\mathbf{A}}$ | The relaxed (continuous) adjacency matrix after Gumbel-Softmax. |
| $\tau$ | The temperature hyperparameter for Gumbel-Softmax. |
| $B$ | The batch size. |
| $k^{(b)}$ | The expected number of edges for the $b$-th graph in a batch. |
| $L$ | The number of layers in the GNN backbone. |
| $\rho(\cdot)$ | A permutation-invariant readout function (e.g., mean, sum). |
| $\mathbf{e} \in \mathbb{R}^{d_e}$ | The graph embedding vector produced by the readout function. |
| *Training Objectives* | |
| $\theta(\phi)$ | The learnable parameters of the GNN, conditioned on resolution $\phi$. |
| $\hat{\mathbf{y}}_i$ | The predicted class logits for sample $i$. |
| $\mathcal{L}_1$ | The label-aligned cross-entropy loss. |
| $\mathcal{L}_2$ | The subject identity contrastive loss. |
| $\tau_{cl}$ | The temperature hyperparameter for the contrastive loss. |
| $\mathcal{R}$ | The $\ell_1$ sparsity regularization term on $\hat{\mathbf{A}}$. |
| $\alpha, \beta$ | Weighting coefficients for the contrastive loss and sparsity term. |
| $\mathcal{L}$ | The total objective function for GCM. |
| *Theoretical Analysis* | |
| $\Sigma_{ij}(\omega)$ | The second-order moment (covariance) between signals $X_i(t)$ and $X_j(t + \omega)$. |
| $\kappa_{i_1 \ldots i_k}(\cdot)$ | The $k$-th order cumulant, used to capture high-order dependence. |
| $f_{ij}(\cdot, \cdot)$ | An edge rule for a pairwise network, depending only on two variables. |
| $\mathbf{X}^{(1)}, \mathbf{X}^{(2)}$ | Two time series used to prove the limitation of pairwise models. |
| $\psi(\cdot)$ | A measurable function representing a high-order interaction rule. |
| $g_\star(\cdot)$ | A class-separable mapping function in the proof of **Theorem 2**. |
| $S$ | In proofs, the set of $k$ nodes on which a ground-truth rule depends. |
| $\mathbf{A}^\star$ | In proofs, the oracle graph structure (e.g., a k-clique). |
| $z_v = [X_v(t), e_v]$ | Enhanced feature for node $v$, combining signal and an ID embedding $e_v$. |
| $x_S$ | The set of node features for all nodes in the set $S$, i.e., $\{z_v | v \in S\}$. |

# B  Semantical Uninterchangeability of the Proposed Scenarios

We cast the four resolutions of functional brain network (FBN) learning (*sample*, *subject*, *group*, and *project*) into a formal hierarchical model and prove *the four random FBN variables are mutually non-equivalent in distribution whenever inter-level variance is non-zero*.

**Hierarchical generative model.** Let $\mathcal{G}^{(p)} \sim \mathbb{P}_\theta$ be the **project-level** FBN, $\mathcal{G}_m^{(g)} \sim \mathbb{P}_{\sigma|\theta}\big(\cdot \mid \mathcal{G}^{(p)}\big)$ the $m$-th **group** FBN, $\mathcal{G}_{mn}^{(s)} \sim \mathbb{P}_{\rho|\sigma}\big(\cdot \mid \mathcal{G}_m^{(g)}\big)$ the $n$-th **subject** FBN in group $m$, and $\mathcal{G}_{mnr}^{(d)} \sim \mathbb{P}_{\eta|\rho}\big(\cdot \mid \mathcal{G}_{mn}^{(s)}\big)$ the $r$-th **sample** (d for "dynamic") network. We assume

$$\mathrm{Var}\big[\mathcal{G}_m^{(g)} \mid \mathcal{G}^{(p)}\big] = \sigma_g^2 > 0, \ \mathrm{Var}\big[\mathcal{G}_{mn}^{(s)} \mid \mathcal{G}_m^{(g)}\big] = \sigma_s^2 > 0, \ \mathrm{Var}\big[\mathcal{G}_{mnr}^{(d)} \mid \mathcal{G}_{mn}^{(s)}\big] = \sigma_d^2 > 0. \quad (1)$$

**Lemma 2** (Non-degenerate variance implies distributional gap)**.** *If* $\mathrm{Var}[\mathcal{X} \mid \mathcal{Y}] > 0$ *almost surely, then* $\mathcal{X}$ *and* $\mathcal{Y}$ *are not equal in distribution.*

*Proof.* Equal distribution would imply $\Pr[\mathcal{X} \neq \mathcal{Y}] = 0$, which forces conditional variance to zero—a contradiction. $\qquad\square$

**Theorem 3** (Mutual non-equivalence)**.** *Under Eq. B the four random graphs* $\mathcal{G}^{(d)}$, $\mathcal{G}^{(s)}$, $\mathcal{G}^{(g)}$, *and* $\mathcal{G}^{(p)}$ *are pairwise non-equivalent in distribution; hence they encode genuinely different semantics.*

*Proof.* Apply Lemma 2 successively: $\sigma_d^2 > 0 \Rightarrow \mathcal{G}^{(d)} \overset{d}{\neq} \mathcal{G}^{(s)}$, $\sigma_s^2 > 0 \Rightarrow \mathcal{G}^{(s)} \overset{d}{\neq} \mathcal{G}^{(g)}$, and so on; transitivity yields all six pairwise inequalities. $\qquad\square$

## C   Proof of Theorem 1

*Proof.* By **Lemma 1**, a random vector is fully determined by its second-order statistics if and only if it is multivariate Gaussian; for non-Gaussian vectors, one can match all second-order cumulants while altering higher-order ($\leq 3$) cumulants. Let $\mathbf{X}^{(1)}$ and $\mathbf{X}^{(2)}$ satisfy

$$\Sigma_{ij}^{(1)}(\omega) \ = \ \Sigma_{ij}^{(2)}(\omega), \quad \forall\, i,j,\ \forall\, \omega \in \Omega,$$

but assume there exists some order $k \geq 3$ such that $\kappa_{i_1\cdots i_k}^{(1)}(\cdot) \ \neq \ \kappa_{i_1\cdots i_k}^{(2)}(\cdot)$.

Since each pairwise network generator $A_{ij} = f_{ij}(X_i, X_j)$ depends only on second-order information derivable from the bivariate distribution of $(X_i, X_j)$ over $\Omega$, matching all second-order moments implies $f_{ij}[\mathbf{X}^{(1)}] \ = \ f_{ij}[\mathbf{X}^{(2)}], \quad \forall\, i,j$. Hence the entire adjacency matrix satisfies $\mathbf{A}^{(1)} = \mathbf{A}^{(2)}$. Meanwhile, the deliberate mismatch in higher-order cumulants guarantees $\mathbf{X}^{(1)} \overset{\mathrm{law}}{\neq} \mathbf{X}^{(2)}$. $\qquad\square$

**Connection to Poisson Graphical Models.** Pairwise log–linear Poisson graphical models [9] fall into the edge-rule template above (edges encode only $\mathbb{E}[X_i X_j]$). Hence Theorem 1 applies verbatim: they too are blind to high-order spike-count synchrony.

**Constructive Example** Consider $d = 3$ binary processes, $X_1(t), X_2(t) \overset{\text{i.i.d.}}{\sim} \mathrm{Bern}(1/2)$, and define

$$X_3(t) \ = \ X_1(t) \ \oplus \ X_2(t) \qquad (\text{XOR in } \{0,1\}). \quad (2)$$

All pairwise covariances vanish, yet the third-order cumulant

$$\kappa_{123} \ = \ \mathbb{E}\big[(2X_1 - 1)(2X_2 - 1)(2X_3 - 1)\big] \ = \ -1 \quad (3)$$

is non-zero. Any covariance-based network therefore returns an empty graph, missing the genuine ternary dependency.

## D   Proof of Theorem 2

We provide a self-contained argument that **GCM** can approximate any $k$-th order decision rule ($k \geq 3$) up to arbitrarily small risk. All statements hold for *finite*[3] second-moment stochastic processes.

---

[3]The probability space of $\mathbf{X}(t)$ is assumed to have finite second moment, a standard condition in neuro-signal modelling.

## D.1 Pre-liminaries and Notation

- $S = \{i_1, \ldots, i_k\}$ denotes the $k$ ROIs on which the ground-truth rule depends.
- $\mathbf{A}^\star \in \{0, 1\}^{N \times N}$ is the $k$-clique connecting $S$.
- $\mathcal{G}_\phi^\tau$ is the random graph generated by the Concrete reparameterisation (§ 3.1) at temperature $\tau$.
- Each node $v$ is equipped with $\mathbf{z}_v = [X_v(t), \mathbf{e}_v] \in \mathbb{R}^{d_x + d_{\mathrm{ID}}}$, where $\mathbf{e}_v$ is a *unique, fixed* ID embedding (one-hot or random orthogonal vector).
- $\mathbf{x}_S$ denotes the set of node features for the nodes in the set $S$, i.e., $\{\mathbf{z}_v | v \in S\}$.
- $L = \max\{1, \mathrm{diam}(\mathbf{A}^\star)\}$ (for a clique $L = 1$).

## D.2 Finite-Constant Approximation of the Oracle Graph

**Lemma 3** (Finite-$K$ Concrete Approximation). *For any $\delta > 0$ and $\xi > 0$ there exist a finite constant $K = K(\delta, \xi)$, parameters $\phi \equiv \{\hat{A}_{ij} \in \{\pm K\}\}$, and a temperature $\tau < \tau_{\delta, \xi}$ such that*

$$\Pr\Big[\big\|\mathcal{G}_\phi^\tau - \mathbf{A}^\star\big\|_\infty < \delta\Big] \geq 1 - \xi. \tag{4}$$

*Proof.* Because the Gumbel noise $\mathbf{M}_{ij} = -\log(-\log u)$ is almost surely finite, choose $K \gg 1$ so that $\sigma\big((K + \max \mathbf{M}_{ij})/\tau\big) > 1 - \delta$ and $\sigma\big((-K + \min \mathbf{M}_{ij})/\tau\big) < \delta$ with probability $1 - \xi$. $\square$

## D.3 Permutation-Invariant Injection via ID-Tagged DeepSets

**Lemma 4** (ID-Tagged DeepSets Injection). *Let $\phi : \mathbb{R}^{d_x + d_{\mathrm{ID}}} \to \mathbb{R}^{d_\phi}$ be injective and continuous. Define*

$$\mathbf{h} = \sum_{v \in S} \phi(\mathbf{z}_v), \quad \mathbf{z}_v = [X_v(t), \mathbf{e}_v]. \tag{5}$$

*Then the mapping $\mathcal{M} : \{\mathbf{x}_S\} \mapsto \mathbf{h}$ is injective on all finite multisets of node features.*

*Proof.* Distinct nodes carry distinct ID embeddings $\mathbf{e}_v$; hence $\phi(\mathbf{z}_{v_1}) \neq \phi(\mathbf{z}_{v_2})$ whenever $v_1 \neq v_2$. Sum aggregation therefore preserves multiset identity. $\square$

## D.4 Truncated Universal Approximation

**Lemma 5** (Probabilistic UA on Unbounded Domain). *For any $\eta > 0$ there exist $T > 0$ and an MLP $\rho_\eta$ such that*

$$\Pr\big[\|\mathbf{x}_S\|_\infty > T\big] < \eta, \tag{6}$$

$$\big|\rho_\eta(\mathbf{h}) - \psi(\mathbf{x}_S)\big| < \eta, \quad \forall \mathbf{h} \in \Big\{\sum_{v \in S} \phi(\mathbf{z}_v) : \|\mathbf{x}_S\|_\infty \leq T\Big\}. \tag{7}$$

*Proof.* Choose $T$ so that (6) holds (Markov's inequality suffices by finite variance). The image of the compact cube $[-T, T]^k$ under the continuous sum (5) is compact. Classical UA theorems guarantee an MLP $\rho_\eta$ that attains (7). $\square$

## D.5 Main Theorem

**Theorem 4** (Higher-Order Expressivity of GCM). *Under the setting above, for any $\varepsilon > 0$ there exist parameters $(\phi^\varepsilon, \theta^\varepsilon)$, temperature $\tau^\varepsilon$, and depth $L$ such that the GCM classifier satisfies*

$$\Pr\big[f_{\theta^\varepsilon, \phi^\varepsilon}(\mathbf{X}(t)) \neq y\big] \leq \varepsilon.$$

*Proof.* Allocate the error budget $\varepsilon = \xi + \eta + \eta_{\mathrm{trunc}}$ with $\xi = \eta = \eta_{\mathrm{trunc}} = \varepsilon/3$.

**Adjacency Realisation.** Apply Lemma 3 with $(\delta, \xi)$ to obtain $(\phi^\varepsilon, \tau^\varepsilon)$ such that adversarial graph mismatch occurs with probability $\xi$.

**One-Layer Aggregation.** Run *one* message-passing layer that computes $\mathbf{h} = \sum_{v \in S} \phi(\mathbf{z}_v)$ ((5)). By Lemma 4, $\mathbf{h}$ injectively encodes $\mathbf{x}_S$.

**Readout Approximation.** Invoke Lemma 5 with $\eta$ to pick $T$ and MLP $\rho_\eta$ satisfying (6)–(7). Define $\tilde{\psi} = \rho_\eta \circ \sum_{v \in S} \phi(\mathbf{z}_v)$ and set a final linear head so that $g_\star \circ \tilde{\psi}$ matches the class labels.

**Risk Upper Bound.** Mis-classification can occur from (i) wrong adjacency ($\xi$), (ii) UA error ($\eta$), or (iii) truncated tail mass ($\eta_{\text{trunc}}$). Summation gives risk $\leq \xi + \eta + \eta_{\text{trunc}} = \varepsilon$. □

*Remark* 1. When $k = 2$ the oracle graph reduces to a single edge, and the above construction collapses to conventional pairwise modelling. For $k \geq 3$, joint gradient updates over all $\hat{\mathbf{A}}$ entries allow GCM to synthesise $\mathbf{A}^\star$—thereby *escaping* the impossibility bound for static pairwise networks established in §2.

# E  Pseudo-code

In this section, we provide the pseudo-code of the proposed framework **GCM** and **BBA**.

---

**Algorithm 1 GCM**

---

**Input:** Samples $\{\mathcal{X}\}$, Level Mapping $\phi$, Graph Numbers $t$, Learning Rate $\eta$, Coefficient $\alpha$, $\beta$
**Output:** Predicted labels $\{\hat{\mathcal{Y}}\}$, Learned graphs $\{\mathcal{G}\}$
Init $\{\hat{\mathcal{A}}\} \sim \mathcal{U}(0, 1)$), Other Trainable Parameters $\theta$
**while** converge **do**
  **for all** $\mathbf{X}_i$ in $\{\mathcal{X}\}$ **do**
    Compute $\tilde{\mathbf{A}}$ according to $\tilde{\mathbf{A}} = \sigma\left(\frac{\mathbf{A}+\mathbf{M}}{\tau}\right)$
    Compute $\tilde{\mathbf{E}}$ for $\tilde{\mathbf{A}}$ according to $\tilde{E} = \frac{1}{2}\sum_{i=1}^{N}\sum_{j \neq i} \tilde{\mathbf{A}}_{ij}$
    Binarize a batch of $\tilde{\mathbf{A}}$ using $\tilde{\mathbf{E}}$ using **BBA**
    Pair $\mathbf{X}_i$ with $\mathcal{G}_{\phi(\mathbf{X}_i)}$ using $\phi(\cdot)$
    Compute Hidden Layers$\mathbf{H}_i^k = \text{GNN}(\mathbf{X}_i, \mathcal{G}_i)$
    Compute $\mathbf{e}_i$ using ReadOut
    Predict Label $\hat{y}_i$ according to $\hat{y}_i = \text{softmax}\left(\mathbf{W}^\top \mathbf{e}_i + \mathbf{b}\right)$
    Compute $\mathcal{L}_1$ according to $\mathcal{L}_1 = -\sum_i y_i \log(\hat{y}_i)$
    Compute $\mathcal{L}_2 = -\frac{1}{|\mathcal{P}_{\text{pos}}|}\sum_{(i,j)\in\mathcal{P}_{\text{pos}}} \log \sigma\left(\frac{\mathbf{e}_i^\top \mathbf{e}_j}{\tau_{cl}}\right) - \frac{1}{|\mathcal{P}_{\text{neg}}|}\sum_{(i,k)\in\mathcal{P}_{\text{neg}}} \log \sigma\left(-\frac{\mathbf{e}_i^\top \mathbf{e}_k}{\tau_{cl}}\right)$
    Compute $\mathcal{R}$ according to $\mathbf{A} = \sigma\left(\hat{\mathbf{A}} + \hat{\mathbf{A}}^\top\right)$
    $\mathcal{L} = \mathcal{L}_1 + \alpha * \mathcal{L}_2 + \beta * \mathcal{R}$
    Update $\mathbf{A}$ by $\hat{\mathbf{A}} := \hat{\mathbf{A}} - \eta \frac{\partial \mathcal{L}}{\partial \tilde{\mathbf{A}}} \frac{\partial \tilde{\mathbf{A}}}{\partial \mathbf{A}} \frac{\partial \mathbf{A}}{\partial \hat{\mathbf{A}}}$
    Update $\theta$ by $\theta := \theta - \eta \,\mathbb{E}_\tau\left[\frac{\partial \mathcal{L}}{\partial \hat{\mathbf{A}}} \frac{\partial \hat{\mathbf{A}}}{\partial \theta}\right]$
  **end for**
**end while**

---

---

**Algorithm 2 BBA**

---

1: **Input:** A batch of adjacency matrices $\mathbf{A} \in \mathbb{R}^{B \times N \times N}$, Thresholds $\mathbf{k} \in \mathbb{R}^B$ consists of the expected edge numbers $k^{(b)}$ of $b$-th adjacency matrix $\mathbf{A}^{(b)} \in \mathbb{R}^{N \times N}$ in the batch
2: **Output:** Batch of network structures $\{\mathcal{G}\}_{i=1}^B$
3: Transform $\mathbf{A}^{(b)}$ into $\mathbf{q}^{(b)} \in \mathbb{R}^{N^2}$
4: Construct $\mathbf{M}$ using $\mathbf{m}_j^{(b)} = \begin{cases} 1 & \text{if } j \in \{\mathbf{t}_1^{(b)}, \ldots, \mathbf{t}_{k^{(b)}}^{(b)}\}, \\ 0 & \text{otherwise.} \end{cases}$
5: Obtain $\mathring{\mathbf{Q}}$ by $\mathring{\mathbf{Q}}^{(b)} = \mathbf{M} \odot \mathbf{Q}$
6: Obtain $\{\mathcal{G}\}_{i=1}^B$ by reshape $\mathring{\mathbf{Q}}$ to the shape of $\mathbf{A}$

---

Table A2: Impact of Initial Structure on Classic Methods (Accuracy$_{\pm Std}$)

| Method | DynHCP$_{Activity}$ | | DynHCP$_{Age}$ | | DynHCP$_{Gender}$ | | CogState | | SLIM | |
| --- | --- | --- | --- | --- | --- | --- | --- | --- | --- | --- |
| | Intra | Inter | Intra | Inter | Intra | Inter | Intra | Inter | Intra | Inter |
| GCN | $0.774_{\pm0.022}$ | $0.736_{\pm0.068}$ | $0.423_{\pm0.022}$ | $0.455_{\pm0.053}$ | $0.650_{\pm0.017}$ | $0.543_{\pm0.045}$ | $0.290_{\pm0.023}$ | $0.303_{\pm0.014}$ | $0.552_{\pm0.012}$ | $0.543_{\pm0.072}$ |
| SAGE | $0.791_{\pm0.005}$ | $0.794_{\pm0.032}$ | $0.464_{\pm0.019}$ | $0.411_{\pm0.034}$ | $0.633_{\pm0.025}$ | $0.594_{\pm0.060}$ | $0.334_{\pm0.012}$ | $0.301_{\pm0.022}$ | $0.657_{\pm0.013}$ | $0.596_{\pm0.053}$ |
| GAT | $0.140_{\pm0.000}$ | $0.138_{\pm0.000}$ | $0.212_{\pm0.000}$ | $0.202_{\pm0.004}$ | $0.542_{\pm0.004}$ | $0.532_{\pm0.004}$ | $0.296_{\pm0.004}$ | $0.305_{\pm0.009}$ | $0.622_{\pm0.029}$ | $0.602_{\pm0.018}$ |
| GIN | $0.480_{\pm0.062}$ | $0.495_{\pm0.058}$ | $0.449_{\pm0.013}$ | $0.401_{\pm0.065}$ | $0.572_{\pm0.012}$ | $0.565_{\pm0.030}$ | $0.308_{\pm0.017}$ | $0.280_{\pm0.023}$ | $0.534_{\pm0.023}$ | $0.465_{\pm0.040}$ |
| GCN$_{w/o\ struc}$ | $0.669_{\pm0.003}$ | $0.612_{\pm0.083}$ | $0.618_{\pm0.023}$↑ | $0.408_{\pm0.027}$ | $0.744_{\pm0.032}$↑ | $0.571_{\pm0.082}$↑ | $0.267_{\pm0.015}$ | $0.269_{\pm0.013}$ | $0.547_{\pm0.018}$ | $0.560_{\pm0.045}$↑ |
| SAGE$_{w/o\ struc}$ | $0.874_{\pm0.012}$↑ | $0.823_{\pm0.021}$↑ | $0.604_{\pm0.011}$↑ | $0.410_{\pm0.035}$ | $0.783_{\pm0.006}$↑ | $0.656_{\pm0.037}$↑ | $0.322_{\pm0.027}$ | $0.326_{\pm0.025}$↑ | $0.638_{\pm0.030}$ | $0.613_{\pm0.012}$↑ |
| GAT$_{w/o\ struc}$ | $0.940_{\pm0.022}$↑ | $0.847_{\pm0.062}$↑ | $0.826_{\pm0.007}$↑ | $0.455_{\pm0.027}$↑ | $0.894_{\pm0.012}$↑ | $0.643_{\pm0.047}$↑ | $0.210_{\pm0.003}$ | $0.255_{\pm0.004}$ | $0.606_{\pm0.010}$ | $0.574_{\pm0.025}$ |
| GIN$_{w/o\ struc}$ | $0.145_{\pm0.012}$ | $0.152_{\pm0.018}$ | $0.451_{\pm0.015}$↑ | $0.442_{\pm0.003}$↑ | $0.545_{\pm0.007}$ | $0.476_{\pm0.026}$ | $0.238_{\pm0.013}$ | $0.247_{\pm0.016}$ | $0.529_{\pm0.027}$ | $0.528_{\pm0.007}$↑ |

To highlight the limitations of pairwise methods, we first report our experiments conducted on GNNs in Table A2. We compared the performance of directly inputting a Pearson-based FBN versus using a fully connected network (FCN) as input for the FBN. The results are marked with a red upward arrow to highlight instances where the predictive performance improved after the replacement. Half of the predictions showed varying degrees of improvement, demonstrating that Pearson-based FBNs, as a representative pairwise approach, fail to fully capture complex relationships in brain activity. Furthermore, using static FCNs revealed their limitations in capturing the structural information inherent in brain networks. These findings emphasize the need for FBN structure learning, which allows for the discovery of richer, more context-sensitive representations of brain activity. We conducted same experiments with GSL and BGI SOTAs for further validation, and the corresponding results are provided in Table A3. Again, over half of these SOTAs perform better without relying on original input structures, further underscoring the importance of exploring more reliable FBN construction techniques.

Table A3: Original vs. w/o Structure Settings (Accuracy$_{\pm Std}$)

| Method | DynHCP$_{Activity}$ | | DynHCP$_{Age}$ | | DynHCP$_{Gender}$ | | CogState | | SLIM | |
| --- | --- | --- | --- | --- | --- | --- | --- | --- | --- | --- |
| | Intra | Inter | Intra | Inter | Intra | Inter | Intra | Inter | Intra | Inter |
| VIB-GSL | $0.925_{\pm0.006}$ | $0.772_{\pm0.017}$ | $0.600_{\pm0.060}$ | $0.429_{\pm0.031}$ | $0.781_{\pm0.029}$ | $0.574_{\pm0.034}$ | $0.313_{\pm0.014}$ | $0.303_{\pm0.010}$ | $0.529_{\pm0.007}$ | $0.537_{\pm0.013}$ |
| | $0.930_{\pm0.006}$↑ | $0.769_{\pm0.012}$ | $0.568_{\pm0.068}$ | $0.435_{\pm0.035}$↑ | $0.791_{\pm0.015}$↑ | $0.596_{\pm0.043}$↑ | $0.315_{\pm0.010}$↑ | $0.301_{\pm0.008}$ | $0.512_{\pm0.012}$ | $0.522_{\pm0.009}$ |
| HGP-SL | $0.715_{\pm0.010}$ | $0.647_{\pm0.053}$ | $0.529_{\pm0.012}$ | $0.404_{\pm0.022}$ | $0.766_{\pm0.006}$ | $0.612_{\pm0.003}$ | $0.286_{\pm0.005}$ | $0.270_{\pm0.008}$ | $0.556_{\pm0.009}$ | $0.552_{\pm0.009}$ |
| | $0.909_{\pm0.013}$↑ | $0.817_{\pm0.017}$↑ | $0.712_{\pm0.035}$↑ | $0.371_{\pm0.014}$ | $0.852_{\pm0.013}$↑ | $0.618_{\pm0.019}$↑ | $0.336_{\pm0.004}$↑ | $0.330_{\pm0.006}$↑ | $0.522_{\pm0.010}$ | $0.504_{\pm0.009}$ |
| IBGNN | $0.908_{\pm0.004}$ | $0.586_{\pm0.278}$ | $0.768_{\pm0.025}$ | $0.397_{\pm0.008}$ | $0.858_{\pm0.027}$ | $0.591_{\pm0.025}$ | $0.343_{\pm0.008}$ | $0.300_{\pm0.012}$ | $0.648_{\pm0.031}$ | $0.617_{\pm0.027}$ |
| | $0.842_{\pm0.056}$ | $0.804_{\pm0.029}$↑ | $0.707_{\pm0.032}$ | $0.396_{\pm0.015}$ | $0.847_{\pm0.044}$ | $0.619_{\pm0.039}$↑ | $0.345_{\pm0.023}$↑ | $0.318_{\pm0.016}$↑ | $0.656_{\pm0.016}$↑ | $0.620_{\pm0.015}$↑ |
| DGCL | $0.831_{\pm0.013}$ | $0.631_{\pm0.017}$ | $0.758_{\pm0.037}$ | $0.392_{\pm0.033}$ | $0.860_{\pm0.043}$ | $0.615_{\pm0.017}$ | $0.316_{\pm0.011}$ | $0.275_{\pm0.007}$ | $0.552_{\pm0.009}$ | $0.539_{\pm0.017}$ |
| | $0.845_{\pm0.072}$↑ | $0.712_{\pm0.043}$↑ | $0.787_{\pm0.023}$↑ | $0.406_{\pm0.040}$↑ | $0.842_{\pm0.057}$ | $0.588_{\pm0.033}$ | $0.319_{\pm0.011}$↑ | $0.297_{\pm0.007}$↑ | $0.548_{\pm0.022}$ | $0.518_{\pm0.007}$ |
| IC-GNN | $0.878_{\pm0.053}$ | $0.744_{\pm0.092}$ | $0.678_{\pm0.105}$ | $0.425_{\pm0.024}$ | $0.851_{\pm0.065}$ | $0.658_{\pm0.067}$ | $0.318_{\pm0.077}$ | $0.313_{\pm0.053}$ | $0.638_{\pm0.103}$ | $0.614_{\pm0.072}$ |
| | $0.881_{\pm0.024}$↑ | $0.767_{\pm0.042}$↑ | $0.653_{\pm0.046}$ | $0.433_{\pm0.021}$↑ | $0.851_{\pm0.033}$ | $0.652_{\pm0.029}$ | $0.321_{\pm0.035}$↑ | $0.314_{\pm0.028}$↑ | $0.642_{\pm0.053}$↑ | $0.634_{\pm0.022}$↑ |
| BrainIB | $0.926_{\pm0.043}$ | $0.839_{\pm0.018}$ | $0.818_{\pm0.035}$ | $0.445_{\pm0.019}$ | $0.873_{\pm0.011}$ | $0.662_{\pm0.020}$ | $0.332_{\pm0.013}$ | $0.325_{\pm0.021}$ | $0.644_{\pm0.022}$ | $0.631_{\pm0.012}$ |
| | $0.947_{\pm0.015}$↑ | $0.843_{\pm0.011}$↑ | $0.808_{\pm0.015}$ | $0.453_{\pm0.017}$↑ | $0.853_{\pm0.009}$ | $0.644_{\pm0.010}$ | $0.346_{\pm0.008}$↑ | $0.333_{\pm0.014}$↑ | $0.652_{\pm0.019}$↑ | $0.640_{\pm0.018}$↑ |

## G  Structure Learning Results Using Additional Metrics

Table A4: Comprehensive Performance Metrics (SEN, SPE, AUC)

| Method | DynHCP$_{Activity}$ Intra SEN | SPE | AUC | Inter SEN | SPE | AUC | DynHCP$_{Age}$ Intra SEN | SPE | AUC | Inter SEN | SPE | AUC | DynHCP$_{Gender}$ Intra SEN | SPE | AUC | Inter SEN | SPE | AUC | CogState Intra SEN | SPE | AUC | Inter SEN | SPE | AUC | SLIM Intra SEN | SPE | AUC | Inter SEN | SPE | AUC |
| --- | --- | --- | --- | --- | --- | --- | --- | --- | --- | --- | --- | --- | --- | --- | --- | --- | --- | --- | --- | --- | --- | --- | --- | --- | --- | --- | --- | --- | --- | --- |
| *— Traditional (Sample-Level) —* | | | | | | | | | | | | | | | | | | | | | | | | | | | | | | |
| LR | 0.941 | 0.937 | 0.978 | 0.823 | 0.819 | 0.890 | 0.811 | 0.807 | 0.885 | 0.360 | 0.354 | 0.685 | 0.880 | 0.878 | 0.929 | 0.627 | 0.623 | 0.680 | 0.279 | 0.273 | 0.610 | 0.221 | 0.215 | 0.550 | 0.525 | 0.517 | 0.575 | 0.542 | 0.536 | 0.590 |
| RF | 0.947 | 0.943 | 0.984 | 0.828 | 0.822 | 0.895 | 0.815 | 0.809 | 0.888 | 0.412 | 0.406 | 0.720 | 0.826 | 0.822 | 0.881 | 0.631 | 0.627 | 0.685 | 0.291 | 0.285 | 0.620 | 0.268 | 0.260 | 0.595 | 0.638 | 0.632 | 0.695 | 0.629 | 0.623 | 0.688 |
| XGBoost | 0.961 | 0.957 | 0.990 | 0.835 | 0.829 | 0.900 | 0.746 | 0.740 | 0.820 | 0.391 | 0.385 | 0.705 | 0.796 | 0.858 | 0.816 | 0.613 | 0.609 | 0.670 | 0.310 | 0.304 | 0.635 | 0.285 | 0.279 | 0.610 | 0.626 | 0.620 | 0.685 | 0.613 | 0.607 | 0.670 |
| SVM | 0.954 | 0.950 | 0.987 | 0.845 | 0.839 | 0.910 | 0.830 | 0.824 | 0.898 | 0.398 | 0.390 | 0.710 | 0.885 | 0.879 | 0.933 | 0.634 | 0.628 | 0.690 | 0.324 | 0.318 | 0.645 | 0.273 | 0.267 | 0.600 | 0.570 | 0.564 | 0.620 | 0.557 | 0.551 | 0.610 |
| MLP | 0.953 | 0.949 | 0.986 | 0.849 | 0.845 | 0.914 | 0.791 | 0.785 | 0.855 | 0.375 | 0.369 | 0.695 | 0.906 | 0.900 | 0.950 | 0.646 | 0.640 | 0.700 | 0.379 | 0.373 | 0.702 | 0.276 | 0.270 | 0.605 | 0.521 | 0.515 | 0.570 | 0.552 | 0.546 | 0.605 |
| *— GNN (Sample-Level) —* | | | | | | | | | | | | | | | | | | | | | | | | | | | | | | |
| GCN | 0.778 | 0.770 | 0.845 | 0.739 | 0.733 | 0.810 | 0.426 | 0.420 | 0.730 | 0.458 | 0.452 | 0.735 | 0.653 | 0.647 | 0.710 | 0.546 | 0.540 | 0.600 | 0.293 | 0.287 | 0.622 | 0.306 | 0.300 | 0.630 | 0.555 | 0.549 | 0.610 | 0.546 | 0.540 | 0.600 |
| SAGE | 0.795 | 0.787 | 0.860 | 0.797 | 0.791 | 0.865 | 0.467 | 0.461 | 0.760 | 0.414 | 0.408 | 0.725 | 0.636 | 0.630 | 0.690 | 0.597 | 0.591 | 0.650 | 0.337 | 0.331 | 0.658 | 0.304 | 0.298 | 0.628 | 0.660 | 0.654 | 0.715 | 0.599 | 0.593 | 0.655 |
| GAT | 0.142 | 0.138 | 0.500 | 0.140 | 0.136 | 0.500 | 0.215 | 0.209 | 0.550 | 0.205 | 0.199 | 0.540 | 0.545 | 0.539 | 0.600 | 0.535 | 0.529 | 0.590 | 0.299 | 0.293 | 0.625 | 0.308 | 0.302 | 0.632 | 0.625 | 0.619 | 0.680 | 0.605 | 0.599 | 0.660 |
| GIN | 0.484 | 0.476 | 0.770 | 0.499 | 0.491 | 0.780 | 0.452 | 0.446 | 0.750 | 0.404 | 0.398 | 0.715 | 0.575 | 0.569 | 0.630 | 0.568 | 0.562 | 0.620 | 0.311 | 0.305 | 0.635 | 0.283 | 0.277 | 0.610 | 0.537 | 0.531 | 0.590 | 0.468 | 0.462 | 0.530 |
| *— GSL & BGI SOTAs (Sample-Level) —* | | | | | | | | | | | | | | | | | | | | | | | | | | | | | | |
| VIB-GSL | 0.927 | 0.923 | 0.970 | 0.775 | 0.769 | 0.845 | 0.603 | 0.597 | 0.820 | 0.432 | 0.426 | 0.735 | 0.784 | 0.778 | 0.845 | 0.577 | 0.571 | 0.630 | 0.316 | 0.310 | 0.640 | 0.306 | 0.300 | 0.630 | 0.532 | 0.526 | 0.585 | 0.540 | 0.534 | 0.590 |
| HGP-SL | 0.719 | 0.711 | 0.810 | 0.650 | 0.644 | 0.710 | 0.532 | 0.526 | 0.790 | 0.445 | 0.439 | 0.715 | 0.790 | 0.785 | 0.830 | 0.615 | 0.609 | 0.670 | 0.289 | 0.283 | 0.620 | 0.273 | 0.270 | 0.600 | 0.559 | 0.553 | 0.615 | 0.555 | 0.549 | 0.610 |
| DGCL | 0.834 | 0.828 | 0.900 | 0.634 | 0.628 | 0.690 | 0.761 | 0.755 | 0.830 | 0.395 | 0.389 | 0.710 | 0.863 | 0.857 | 0.920 | 0.618 | 0.612 | 0.675 | 0.319 | 0.313 | 0.640 | 0.278 | 0.272 | 0.605 | 0.555 | 0.549 | 0.610 | 0.542 | 0.536 | 0.595 |
| IBGNN | 0.910 | 0.906 | 0.960 | 0.589 | 0.583 | 0.650 | 0.771 | 0.765 | 0.840 | 0.400 | 0.394 | 0.710 | 0.861 | 0.855 | 0.920 | 0.600 | 0.594 | 0.650 | 0.321 | 0.315 | 0.645 | 0.316 | 0.310 | 0.640 | 0.651 | 0.645 | 0.675 | 0.620 | 0.614 | 0.675 |
| IC-GNN | 0.881 | 0.875 | 0.930 | 0.747 | 0.741 | 0.820 | 0.681 | 0.675 | 0.800 | 0.428 | 0.422 | 0.730 | 0.854 | 0.848 | 0.910 | 0.661 | 0.655 | 0.720 | 0.321 | 0.315 | 0.645 | 0.316 | 0.310 | 0.640 | 0.641 | 0.635 | 0.695 | 0.617 | 0.611 | 0.670 |
| BrainIB | 0.928 | 0.924 | 0.970 | 0.842 | 0.836 | 0.905 | 0.821 | 0.815 | 0.890 | 0.448 | 0.442 | 0.740 | 0.876 | 0.870 | 0.925 | 0.665 | 0.659 | 0.725 | 0.335 | 0.329 | 0.655 | 0.328 | 0.322 | 0.650 | 0.647 | 0.641 | 0.700 | 0.634 | 0.628 | 0.690 |
| *— High-Order SOTAs & Proposed GCM (Sample-Level) —* | | | | | | | | | | | | | | | | | | | | | | | | | | | | | | |
| BNT | 0.976 | 0.972 | 0.998 | 0.842 | 0.836 | 0.908 | 0.948 | 0.942 | 0.985 | 0.389 | 0.383 | 0.705 | 0.979 | 0.973 | 0.995 | 0.681 | 0.675 | 0.735 | 0.399 | 0.393 | 0.720 | 0.340 | 0.334 | 0.665 | 0.556 | 0.550 | 0.610 | 0.532 | 0.526 | 0.580 |
| Hybrid | 0.805 | 0.797 | 0.870 | 0.769 | 0.763 | 0.840 | 0.528 | 0.522 | 0.790 | 0.445 | 0.439 | 0.495 | 0.791 | 0.785 | 0.850 | 0.595 | 0.589 | 0.645 | 0.296 | 0.290 | 0.625 | 0.289 | 0.283 | 0.615 | 0.540 | 0.534 | 0.590 | 0.525 | 0.519 | 0.575 |
| GCM$_{sample}$ | 0.965 | 0.961 | 0.992 | 0.855 | 0.849 | 0.918 | 0.856 | 0.850 | 0.915 | 0.466 | 0.460 | 0.752 | 0.940 | 0.934 | 0.978 | 0.671 | 0.665 | 0.728 | 0.355 | 0.349 | 0.680 | 0.359 | 0.353 | 0.685 | 0.683 | 0.677 | 0.745 | 0.661 | 0.655 | 0.720 |
| *— Subject-Level Models —* | | | | | | | | | | | | | | | | | | | | | | | | | | | | | | |
| IGS | 0.865 | 0.857 | 0.925 | 0.859 | 0.853 | 0.920 | 0.811 | 0.805 | 0.885 | 0.463 | 0.457 | 0.510 | 0.885 | 0.879 | 0.932 | 0.611 | 0.605 | 0.665 | 0.329 | 0.323 | 0.652 | 0.326 | 0.320 | 0.650 | 0.544 | 0.538 | 0.595 | 0.655 | 0.649 | 0.710 |
| GCM$_{subject}$ | 0.963 | 0.965 | 0.994 | 0.860 | 0.854 | 0.920 | 0.892 | 0.886 | 0.942 | 0.476 | 0.470 | 0.525 | 0.942 | 0.936 | 0.980 | 0.673 | 0.667 | 0.732 | 0.416 | 0.410 | 0.740 | 0.367 | 0.361 | 0.695 | 0.684 | 0.678 | 0.745 | 0.677 | 0.671 | 0.735 |
| *— Group-Level Models —* | | | | | | | | | | | | | | | | | | | | | | | | | | | | | | |
| IBGNN+ | 0.769 | 0.761 | 0.835 | 0.456 | 0.450 | 0.505 | 0.789 | 0.783 | 0.850 | 0.383 | 0.377 | 0.420 | 0.884 | 0.878 | 0.930 | 0.599 | 0.593 | 0.650 | 0.359 | 0.353 | 0.680 | 0.318 | 0.312 | 0.642 | 0.670 | 0.664 | 0.725 | 0.647 | 0.641 | 0.700 |
| GCM$_{project}$ | 0.971 | 0.967 | 0.996 | 0.872 | 0.866 | 0.928 | 0.881 | 0.875 | 0.935 | 0.465 | 0.459 | 0.515 | 0.936 | 0.930 | 0.978 | 0.677 | 0.671 | 0.731 | 0.440 | 0.434 | 0.761 | 0.422 | 0.416 | 0.748 | 0.658 | 0.652 | 0.710 | 0.647 | 0.641 | 0.698 |
| *— Project-Level Model —* | | | | | | | | | | | | | | | | | | | | | | | | | | | | | | |
| GCM$_{group}$ | 0.973 | 0.969 | 0.997 | 0.858 | 0.852 | 0.915 | 0.894 | 0.888 | 0.945 | 0.492 | 0.486 | 0.540 | 0.961 | 0.955 | 0.989 | 0.689 | 0.683 | 0.745 | 0.496 | 0.490 | 0.805 | 0.457 | 0.451 | 0.775 | 0.695 | 0.689 | 0.760 | 0.697 | 0.691 | 0.765 |

To provide a more comprehensive analysis, we present the comparative results under various evaluation metrics, and the findings are consistent with those reported in the main text. Overall, our proposed **GCM** framework for four different resolutions rank at the top across the new metrics (SEN, SPE, and AUC), which further validates the effectiveness and robustness of our approach. The analysis again reveals that at the sample-level, although BNT shows exceptionally strong performance in intra-subject tasks, $\text{GCM}_{\text{sample}}$ demonstrates more comprehensive advantages in the more challenging inter-subject generalization tasks. Furthermore, at higher resolutions, $\text{GCM}_{\text{subject}}$ and $\text{GCM}_{\text{group}}$ also exhibit outstanding performance, once again validating the value of our multi-resolution modeling framework.

## H  Sensitivity Analysis

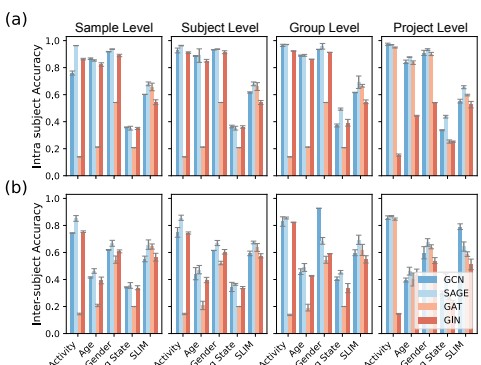

Figure A1: Accuracy of **GCM** using different GNN backbones.

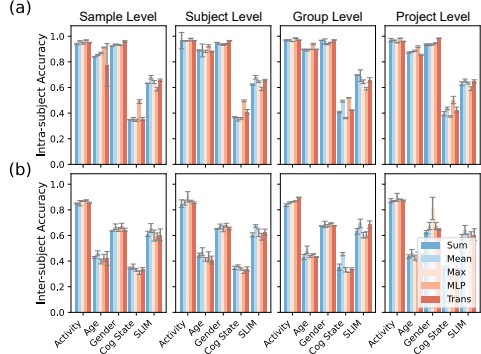

Figure A2: Accuracy of **GCM** using different read out functions.

**Choice of GNN Backbone**    The GNN module serves as the core of **GCM**, modeling local node interactions and extracting global representation of data. Figure A1 presents the performance of different GNN backbones. **SAGE** demonstrates superiority over other architectures under most of conditions, making it the preferred choice for our applications. Notably, further trials, such as using GCN on DynHCP$_{Activity}$ at the *project-level*, indicate potential for higher accuracy.

The readout function determines how node embeddings are aggregated into graph-level representations. We evaluate the performance of various readout functions for **GCM**, as shown in Figure A2. MLP and Transformer are trainable, others are static. The results indicate that a simple mean readout is effective for most conditions. In some cases, trainable functions like MLP or Transformer provide improved precision, suggesting that the choice of readout function should align with specific application requirements.

**Parameter Sensitivity**    Based on the training strategy, we further evaluate the prediction sensitivity to the hyperparameters $\alpha$ and $\beta$. The results are shown in Figure A3. In general, the performance of **GCM** is relatively stable to different choices of these two hyperparameters. However, the results also show that the predictability of **GCM** can be enhanced with a proper coefficient set. For example, higher $\alpha$ and moderate $\beta$ show better predicting accuracy in group level conditions.

## I  Visualization of FBN Structures

We visualize the top-100 weighted edges in male and female FBNs learned by competitors. Subgraph-based methods are omitted because we focus on FBN structure learning for the entire brain rather than sub-FBN extraction. Nodes are represented as colored segments along the circle, edges as curves, and segment widths indicating node degrees. Because our aim is to extract high-order patterns from MTS, and the co-activated brain regions are depicted in the main paper as **Figure 5**, we mainly present the comparison on edge and node distributions of the learned FBNs between baselines. According to the visualizations, the averaged FBN structures of baselines represent more randomness compared to the

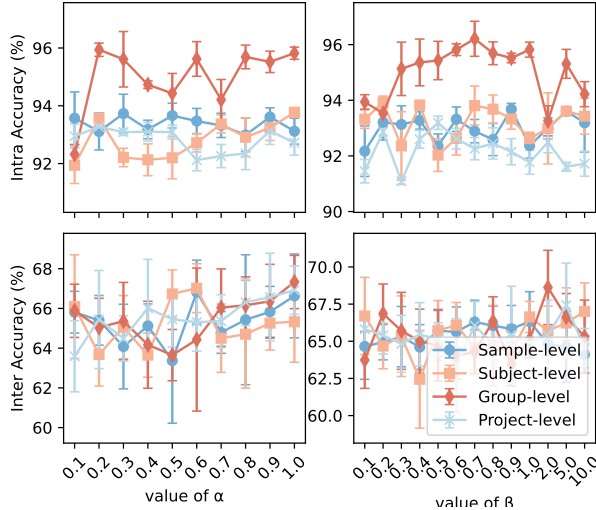

Figure A3: Results using different values of $\alpha$ and $\beta$.

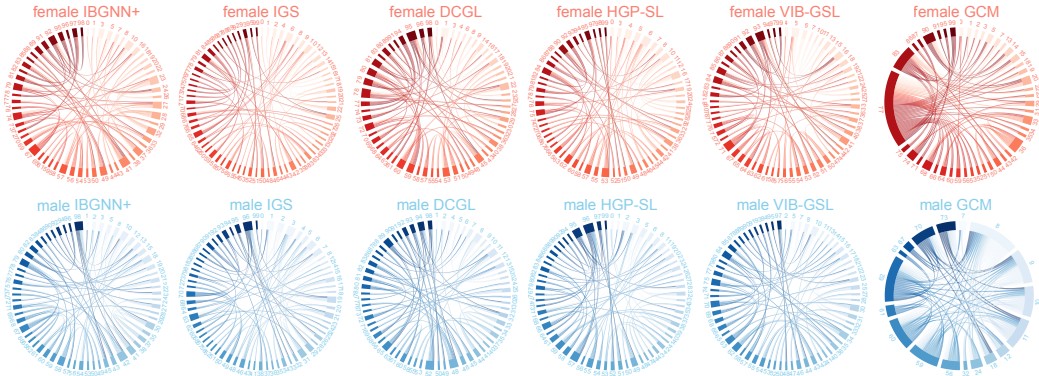

Figure A4: Male (red) and female (blue) FBNs on DynHCP$_{\text{Gender}}$ show top 100 weighted edges obtained by different methods.

structure learned by **GCM**group, emphasizing the inadequacy of simple averaging across research levels. Prominent hub nodes can be observed in the FBNs learned by **GCM**group, further enhancing the result that high-order interactions are not randomized [17]. Notably, **GCM**group reveals distinct patterns: female FBNs have more nodes and fewer hubs than male FBNs, which aligns with the conclusions of the previous study in neuroscience [94]. demonstrating that the learned structures are both meaningful and non-trivial.

## J  Reproduction Settings

The detailed settings of our experiment are provided in this section.

### J.1  Parameter Settings

For the traditional methods, they are implemented using the `scikit-learn` package with their default parameter settings. For the method needing input structures, we computed graph structures by selecting the top-10% entries of the Pearson correlation matrices calculated from samples. We also set a scenario where the input structures are fully connected networks, representing the same condition as our proposed **GCM** where there are no initial structures. We randomly selected 10

samples for each subject on each data set for speeding up, except for SLIM. We shuffled and split each data set into training/validation/test sets according to each task by the ratio of $7 : 1 : 2$.

We set $\tau = 1$ in the Gumbel-softmax, $\tau_{cl} = 1$ in the contrastive loss, $\alpha$ and $\beta$ are selected by grid searching. **SAGE** is set as the backbone for **GCM** in comparison. Common parameter settings are summarized as follows:

- **Learning Rate** is set as $0.001$ for each competitors.
- **Epochs** are set to $1,000$ for each competitors.
- **Early Stop** thresholds are set to $10$ for deep models.
- **Batch Size** is set to $32$ for deep models.
- **Layers** are set as $2$ for deep models.
- **Hidden Dimensions** are set as $128$ for deep models.
- **Output Dimensions** are set as $2$ for deep models.
- **Backbone** is set for GSL and BGS models as default: GIN for VIB-GSL, GCN for HGP-SL, IBGNN and IGS, respectively.
- **Readout** is set as Mean pool in GNN-based methods for fair comparison except for HGP-SL, which is set as HGPSL-Pool by default.
- **Head** is set to $8$ for GAT.
- **Network Density** is set to $10\%$ for methods needing a fixed threshold.
- **The rest of the parameters** are set as default.

## J.2 Environment

All the experiments were performed on a Linux cluster equipped with Intel(R) Xeon(R) Platinum 8358P CPU @ 2.60GHz, Nvidia A800 80G GPU and 520 GB RAM.

## J.3 Data Description and Preprocessing

**Human Connectome Project (HCP)** The Human Connectome Project[4] offers a valuable, publicly accessible neuroimaging dataset that encompasses a comprehensive array of imaging, behavioural, and cognitive data. Our study leverages dynamic networks derived from both resting-state and seven distinct task-based fMRI paradigms. Thus, there are three distinct predicting objectives according to the dataset, namely *Activity*, *Age*, and *Gender*, respectively. We use an open-source version standardized and processed by NeuroGraph[5]. The dynamic graphs are generated via a sliding window approach, characterized as a window length of 50, a stride of 3, and a dynamic length of 150, which is applied to the preprocessed time series.

**Cognitive State** Cognitive State[6] dataset (Cog State) collects EEG data from a group of 60 healthy undergraduate subjects, resulting in a total of 900 records. The primary objective is to validate the reproducibility of EEG metrics across different cognitive states. The EEG records are categorized into five cognitive states: Eyes Open (EO), Eyes Closed (EC), Math (Ma), Memory (Me), and Music (Mu). Each EEG record, originally sampled at a resolution of 500 Hz over a period of 5 minutes, thus comprises substantial $150,000$ data points across 61 channels. These channels adhere to the extended 10-20 system, a widely recognized standard in EEG studies. We initially down-sampled the data to 100 Hz using EEGLAB[7], a comprehensive MATLAB[8] plug-in designed for EEG data analysis. The down-sampling process resulted in a manageable $61 \times 30,000$ matrix per record. Subsequently, we employed a non-overlapping sliding window technique to segment the data into manageable 3-second fragments. This segmentation yielded approximately 100 fragments per subject, each represented as a $61 \times 300$ matrix and retained its corresponding cognitive state label.

---

[4] https://www.humanconnectome.org
[5] anwar-said.github.io/anwarsaid/neurograph.html
[6] openneuro.org/datasets/ds004148
[7] sccn.ucsd.edu/eeglab/index.php
[8] mathworks.com/products/matlab.html

**SLIM**  The SLIM[9] dataset includes rest-state fMRI data from a cohort of $1,045$ subjects, using gender as the labels for analysis. We screened out the incomplete data and retained $541$ subjects. We constructed brain functional networks based on the Power 264 atlas, a well-established brain parcellation template. We remove $28$ undefined regions resulting in 236 ROIs as the nodes of FBN. The methodology for this construction is twofold:

1. Using DPABI[10], a MATLAB plug-in, we delineate brain regions according to Power's cortical parcellation template, identifying $264$ regions. The time series signal for each region is derived by averaging voxel signals within a 5mm radius sphere.

2. A 30-second, non-overlapping sliding window technique is applied to segment time series into fragments (dimensions: $236 \times 30$). Pearson correlations between brain regions within each window are calculated, forming dynamic connectivity matrices ($236 \times 236$).

## K   Introduction of Baselines

Table A5: Taxonomy of Baselines

| Characteristic | Traditional | GNN | BGI | GSL | **GCM** |
|---|:---:|:---:|:---:|:---:|:---:|
| End-to-End | ✗ | ✓ | ✓ | ✓ | ✓ |
| Structure Learning | ✗ | ✗ | ✓ | ✓ | ✓ |
| Global Constraint | ✗ | ✗ | ✗ | ✗ | ✓ |
| Multi Resolution | ✗ | ✗ | ✗ | ✗ | ✓ |
| Adaptive Threshold | ✗ | ✗ | ✗ | ✗ | ✓ |

To verify the suitability of **GCM** on FBN learning, we choose the following baselines. Property comparison between **GCM** and baselines are summarized in Tabel A5.

**Traditional Models**  To ensure a fair assessment, these methods are implemented using `scikit-learn` package[11].

- **LR**: logistic regression, using a logistic function to model the relationship between features and the target variable.
- **SVM**: Support vector machine, which determines the decision plane through support vectors. We use `RBF` kernel for implementation.
- **RF**: Random Forest, an ensemble method that operates by constructing a multitude of decision trees and outputs the mode of their predictions.
- **XGBoost**: Extreme Gradient Boosting, an efficient implementation of gradient boosting trees that builds models in a sequential, stage-wise fashion.
- **MLP**: feed-forward artificial neural network, features multiple layers of interconnected 'neuron'.

**GNN Models**  To ensure a fair assessment, these models are implemented in a standardized dense form as **GCM** using `PyTorch Geometric`[12].

- **GCN** [74]: graph convolutional network, adapts convolutional processes to graph structures.
- **SAGE** [79]: an inductive learning framework that creates node embeddings by sampling and aggregating features from the nodes' local neighbourhoods.
- **GAT** [80]: graph attention network, utilizes self-attention mechanisms to assign varying degrees of importance to different nodes in a graph.
- **GIN** [81]: graph isomorphic network, aims to match the distinguishing power of the Weisfeiler-Lehman graph isomorphism test.

---

[9] `https://fcon_1000.projects.nitrc.org/indi/retro/southwestuni_qiu_index.html`
[10] `rfmri.org/DPABI`
[11] `scikit-learn.org`
[12] `pytorch-geometric.readthedocs.io/en/latest/`

**GSL SOTAs**    The open-source methods are experimented with using the original code released by their authors.

- **VIB-GSL** [70]: a framework employing the information bottleneck principle to selectively filter out irrelevant information thus enhancing graph representation learning.
- **HGP-SL** [71]: combines structure learning with hierarchical graph pooling in a GNN to capture and preserve vital topological information from graphs.
- **DGCL** [8]: learn FBNs based on pair-wise interactions between features learned by the encoder designed for DTI data. We re-implement it according to the original paper to suit EEG and fMRI data.

**BGI SOTAs**    These methods are experimented with using the original code released by their authors.

- **IBGNN** [82]: constructs FBNs by Pearson correlation and utilizes a two-layer MLP to merge topological information for prediction.
- **IBGNN+** [82]: learns a sparsified mask as a unified filter on the FBNs after IBGNN is trained.
- **IGS** [51]: learns to iteratively remove task-irrelevant edges of FBNs during training for sparsification.
- **IC-GNN** [53]: A Granger causality-inspired graph neural network using subgraph for interpretable brain network-based psychiatric diagnosis.
- **BrainIB** [39]: A mutual information-based framework for interpretable and robust brain network analysis in psychiatric diagnosis.

**Additional SOTA Models**    These models were incorporated as suggested during the review process to further benchmark GCM's performance.

- **BNT** [83]: Brain Network Transformer, a state-of-the-art model that applies a Transformer architecture to brain networks, using attention mechanisms to achieve high predictive performance.
- **Hybrid** [49]: A conceptually related model that learns a fixed quantity of explicit high-order interactions (hyperedges) in an end-to-end manner.

## L   Limitations

The limitations of **GCM** framework are summarized as follow: (1) Though the framework is generalizable to other MTS-based tasks, further validation on diverse modalities (e.g., MEG) and clinical conditions is needed. (2) Although **GCM** captures high-order synchronization and yields interpretable results, the underlying neurobiological mechanisms remain underexplored. (3) While we formalize the semantic distinctions across four modeling resolutions, their joint modeling and integration remain unaddressed. (4) The experiments in this study are conducted on datasets with relatively balanced class distributions. The performance of the contrastive loss, in particular, on highly imbalanced clinical datasets is an important area for future investigation.

## M   Societal Impacts

Given that the research in this paper involves neurological analysis, it could be used as an approach to disease diagnosis. Thus, it is essential to declare the potential negative societal impacts of this study, even though it is currently in the research phase and has not yet been applied in practice. Specifically, in the process of AI-assisted disease diagnosis, erroneous results are inevitable. Such errors can have severe consequences for patients and society. Therefore, in real-world medical diagnostic scenarios, the final decision should always rest with the physician's diagnosis.

