# OpenReview forum: "Beyond Pairwise Connections: Extracting High-Order Functional Brain Network Structures under Global Constraints"
_NeurIPS.cc/2025/Conference — NeurIPS 2025 poster_

### Official Review · Reviewer_vyUr · 2025-06-20

**Clarity:** 3
**Significance:** 4
**Originality:** 4
**Rating:** 5
**Confidence:** 2

**Summary:**

This paper introduces a new framework for functional brain network modelling named Global Constraints oriented Multi-resolution (GCM), considering not only the typical sample-level modelling, but also including other resolutions from subject, group, and project levels. Experimental results showed significant accuracy improvements with regards to 5 datasets and 2 task settings, while also reporting significant reductions in computational time.

**Questions:**

As I said, I believe this paper to be really relevant to the community, however, given the weaknesses I have identified, I can only recommend borderline accept, and I will be happy to reconsider my scores if the authors tackle the 4 weaknesses points I've identified.

With regards to the lack of clarity previously mentioned as Weakness, I left two points here asking for more specific clarifications:
1. Could the authors clarify some key terms used in the paper, namely what they really mean with:

   1.1. Global constraints. This term is used a lot in Introduction as a key concept of the paper, inclusive in the model's name, but I don't believe this is ever really explained. I would have expected this explained in the Methodology section, but it was not.

   1.2. High-order dependencies/structures/interactions/.... Again, used as a key concept of the paper and in the model's name, but never really defined. I am guessing that this term seems to come from previous literature on hypergraphs specified in the paper, but for such a wide audience like the one in neurips, I'd expect that such a key concept would at least be briefly defined in the main paper such that everyone could understand the main parts of the model. It is even claimed in the abstract that "Higher-order dependencies" is theoretically analysed in the paper.

   1.3. FBN structure (learning at 4 levels). The word "structure" is used a lot in the paper in the context of learning/optimisation, and from Section 3, we get a more formal definition that an FBN structure is basically an undirected graph, which I understand can be differently learned at each distinct resolution level. However, how is this structure actually learned in the model for the group- and project-levels? In the Methodology section we can see that the label-aligned loss allows for information at sample-level to be learned in the graph, and the subject identity contrast seems to allow for information at the subject-level to be learned in the corresponding graphs, but nothing is said about the remaining 2 resolutions. In my opinion this gets even more confusing at Section 6. At the beginning of it, it is said that the "FBN structure learned by GCM" can be used as a "backbone" for a GNN classifier. If a structure is a graph, how can it be a backbone, given backbones are typically neural networks? Furthermore, it is not clear what are the "ground-truth" group- and project- graphs/labels, to understand what the performance really means here.

   1.4. Inter-subject and intra-subject classification. In Section 6.1 I do not believe this difference is clear at all.

2. Can the authors clarify what are the baselines referred on line 252, with regards to the GCM_{group} model? From table 3, I do not seem to see any corresponding model for direct comparison (only one model for each subject and project levels)

**Ethical Concerns:**

["NO or VERY MINOR ethics concerns only"]

**Final Justification:**

The rebuttal provided by the authors was extremely useful for me to better understanding some points and better appreciate the work's contributions. All in all I think this is a very interesting work and I changed my score to Accept.

**Limitations:**

Yes. However, I'd recommend the authors to add a reference for the potential negative societal impacts in the Conclusion section. At the moment only the limitations are pointed to in Conclusion.

**Paper Formatting Concerns:**

None.

**Quality:**

4

**Strengths And Weaknesses:**

The topic of learning a graph/matrix from EEG or fMRI data is definitely a (still) on-going challenge in the field, and the fact that this paper introduces a way to effectively move away from the typical pairwise Pearson correlations is very significant to the field, with the framework of 4 resolutions being a novel and creative way to frame this problem, as far as I know. The theoretical proof that pairwise-based models might not be enough to recover all the dependencies from the original timeseries is a bit obvious in the sense that functions like the Pearson correlation are not bijective, but having a theoretical proof about this is definitely a strength of this paper. With regards to the quality of the paper, the experiments section is technically sound in the context of the paper, and it's good to see that the authors used quite a varied number of baselines and datasets, while also using more traditional ML models for a fairer comparison. Finally, it is good that the model claims to have more fundamental explainability capabilities embedded in the modelling, which is important to move these discussions towards a more clinical practice, even though the task chosen was the not-so-interesting male/female binary classification.

I am not familiar with the hypergraphs and hyper networks literature, which means I am not completely sure about some of the claims of this paper. However, I identify three main weaknesses on this work:
1. For the more traditional ML models, I believe that the paper is missing an important inclusion of Random Forests/XGBoost type of models, which I've seen to work better in the neuroscientific contexts when compared to LR and SVM. In this sense, it would be important to indicate what hyperparameters were used here, for instance, what was the kernel type used for SVM?
2. In an applied neuroscientific paper like this, using only accuracy as an evaluation metric is definitely not enough, and one would expect other metrics like sensitivity, specificity, and AUC ROC, even if left in appendix.
3. When making a computational analysis like the one reported in figure 3, it is important to have some measure of variability. Furthermore, the paper does not specify if this calculations are related to training or inference.
4. I do not know whether it's because of my lack of knowledge of the specific hypergraphs literature, but I believe this paper is not clearly written, and I leave some questions about this in the Questions section.


Some small points:
1. I'm guessing "organismal" on line 51 is a typo?
2. On line 161, missing "is" between "It composed".

---

> ### Author Rebuttal · Authors · 2025-07-26
>
> We sincerely thank the you for catching our contribution to the field and the collegial atmosphere you fostered in your review!
>
> Your questions are valuable as they touch upon a core challenge: bridging the objectives of ML and neuroscience. A typical ML paper seeks **prediction accuracy**, while neuroscience uses prediction to learn an **interpretable model** (our FBN). Our work adopts the latter perspective, which we hope clarifies our design choices below.
>
> # Responses to Questions
>
> ## Q1: Clarification of Key Terms
>
> Thank you for carefully highlighting these core terminologies! Our work intersects **network science, deep learning, and neuroscience**, and our terminology may not have been immediately accessible to all audiences. Your feedback provides a valuable opportunity for us to clarify these concepts, fostering a common ground for this exciting, integrated area of research. We elaborate on each term below and will revise the manuscript to improve this accessibility.
>
> ### Q1.1 Global Constraints
>
> Thank you for this crucial point! To give a precise definition:
>
> **Global Constraints** are top-down rules that **guide** how we learn a Functional Brain Network (FBN) structure **from the input MTS data**. Their purpose is to make the full adjacency matrix a single, learnable **output** of our system.
>
> In our work, these constraints **include: 1) an adaptive sparsity based on each FBN's expected edge number, 2) task labels, and 3) subject identity**. We will add this clear definition to the Introduction and detail it in the Methodology (Sec. 5).
>
> ### Q1.2 High-Order Dependencies
>
> Conventionally, an edge is a **first-order** relationship between two nodes. The simultaneous activation of multiple neurons can be captured as **high-order** dependencies within the network, referring to relational patterns that involve more than two nodes and edges.
>
> In particular, neural activations that are influenced by combinations of multiple neurons represent another type of high-order dependency. For example, consider a brain region (C) activates only when its two driving regions (A and B) are in **opposite states** (i.e., one is active and the other is inactive). This interaction can be directly modeled as a high-order XOR relationship. However, pairwise statistics would fail to capture it, as no stable pairwise synchrony exists between any of the three regions.
>
> The crucial distinction lies in _how_ edges are generated. The adjacency matrix of high-order dependencies is learned via global updates, where gradients are applied to the entire structure. This means the resulting edge weights no longer simply encode pairwise (first-order) relationships, but instead reflect interdependent relationships that emerge from the whole-network optimization. We will add this intuitive definition to the Introduction and revise the abstract to better convey "Pairwise interactions cannot fully capture high-oder dependencies. This limitation is theoretically analysed".
>
> ### Q1.3. Clarifying FBN structure learning at group/project-levels and "backbone".
>
> Thank you for this detailed question! It highlights several key areas where our explanation was not sufficiently clear. We apologize for the ambiguity and will address each of your points systematically below.
>
> **1. The FBN "Structure" as a Learnable Adjacency Matrix**
>
> You are absolutely correct that the FBN "structure" is an undirected graph, represented by its adjacency matrix. The key innovation is that the entire $N×N$ adjacency matrix is treated as a holistic, learnable parameter within our framework, which is optimized end-to-end.
>
> **2. How Group- and Project-Level Structures are Learned**
>
> We sincerely apologize for not making this mechanism explicit in the paper. The learning process for all four resolutions relies on the same loss function, but the difference lies in how this learnable adjacency matrix is shared across samples:
>
> - **For Sample-Level**: Each sample $\mathbf{X}_i$ has its own unique FBN adjacency matrix $\mathbf{A}_i$, which is updated only by the loss computed on that sample.
>
> - **For Subject-Level**: All samples from the same subject share one single FBN adjacency matrix $\mathbf{A}_{subject}$. Gradients from all samples belonging to that subject are aggregated to update this shared matrix.
>
> - **For Group-Level**: All samples belonging to the same group (e.g., a specific clinical cohort or, in our case, a task label like 'female') share one single, common FBN adjacency matrix. For instance, when processing samples with the label 'female', the gradients on these samples are used to update the single, shared 'female' group-level graph.
>
> - **For Project-Level**: Similarly, all samples in the entire dataset share one single project-level adjacency matrix. Gradients from all samples, across all batches, contribute to the update of this one global graph.
>
> Essentially, for group and project levels, the adjacency matrix acts as a shared, learnable parameter that accumulates structural information from all relevant samples. We will explicitly add this detailed explanation to Section 5 of the revised manuscript.
>
> **3. "Backbone" Terminology**
>
> The term "Backbone" refers to using the learned FBN structure as the basis for massage passing on GNN. We agree that "backnone" is imprecise in this context. "Computing graph" or "input structure" for the GNN classifier is indeed more accurate. We will correct this terminology.
>
> **4. "Ground-Truth" and Evaluation at Higher Levels**
>
> Thank you for raising this crucial point. Across all four resolutions, the prediction task itself is identical: predicting the label of each sample.
>
> The key difference, which addresses your question about "ground-truth" for higher levels, is the role of this label, or information. For the group-level, the label (e.g., 'female') serves as a mapping mechanism: it directs the framework to associate all samples with the 'female' label to the single, shared 'female' group-level FBN. The gradients from predicting the sample-level labels of all these individual 'female' samples are then aggregated to update this one representative graph.
>
> Therefore, the accuracy score here is an index of the learned FBN's **quality and representativeness**.
>
> This reframing gets to the heart of our paper's contribution. We leverage a standard ML task (classification) not as an end in itself, but as the supervisory signal to achieve a neuroscience goal: learning an interpretable, high-order brain network. We will make this distinction much clearer in the revised manuscript.
>
> We hope this systematic breakdown fully clarifies the learning mechanism at all resolutions. Your questions have helped us identify a significant gap in our explanation, and the revised manuscript will be much stronger for it.
>
> ### Q1.4 Inter- vs Intra-subject Classification
>
> We apologize for the lack of clarity. We will revise Section 6.1 to explicitly state:
>
> - **Intra-subject classification:** Training and testing sets are created _for each subject individually_. This assesses if a model can learn stable, subject-specific patterns over time.
>
> - **Inter-subject classification:** Subjects are split into training and testing groups. This assesses if a model can learn patterns that generalize across different individuals.
>
> ## Q.2 Clarify the baselines and evaluation of GCM$_\text{group}$
>
> Thank you for this sharp observation! You are correct; direct baselines for GCM$_\text{group}$ are absent due to a fundamental paradigm difference.
>
> From a conventional machine learning perspective, our group-level setup could raise concerns about "label leakage," which is precisely why prior methods have typically avoided this modeling approach.
>
> However, as we clarified in our response to Q1.3, our primary goal is not prediction for its own sake, but to learn a generalizable structural representation for a cohort. Learning such cohort-level FBNs is an essential and often unavoidable task in computational neuroscience (e.g., for identifying biomarkers for a clinical group).
>
> Therefore, we deliberately isolated this resolution to propose a valid framework for this important, domain-specific problem. This novelty is why direct competitors are not readily available, a point we will clarify in the revision.
>
> # Addressing Listed Weaknesses
>
> We also appreciate the your detailed summary of specific weaknesses. Below, we respond to each of these concerns in turn.
>
> ## W1. Lack of traditional baselines
>
> Thank you for your insightful advice on strengthening our baseline comparisons. You are correct that Random Forest and XGBoost are powerful classifiers in this domain, and we agree their inclusion will make our empirical evaluation more comprehensive. We will add them to the revised paper and clarify that the SVM used an RBF kernel.
>
> ## W2. On the use of additional evaluation metrics beyond accuracy
>
> This is an excellent suggestion! We will add tables reporting sensitivity, specificity, and AUC-ROC for all relevant experiments in the appendix.
>
> ## W3. Clarifying variability and training/inference cost in Figure 3
>
> Thank you for your detailed review! Figure 3 reports the average **training time**. We will revise the figure to include error bars representing the standard deviation to show variability.
>
> Regarding inference cost, we will clarify in the revision that for GNN-based models like ours, inference is extremely fast (e.g., **on the order of milliseconds per sample**), making it a non-factor for practical application compared to the training cost.
>
> ## Response to seggestion on limitation and societal impacts
>
> Thank you for this thoughtful suggestion! In the revised version, we will add a sentence to the Conclusion section that points to the potential negative societal impacts in Appendix K.
>
> ## Typos and minor issues
>
> Thank you for your careful reading! We will thoroughly proofread the manuscript to eliminate any remaining errors in the revised version.

---

> > ### Comment · Reviewer_vyUr · 2025-08-02
> >
> > I really appreciate the time taken by the authors to clarify all my questions. In this sense, for me the paper makes so much more sense. This is enough for me to increase my scores to an acceptance level, as I don't have any further question/feedback. For transparency, I cannot increase to a maximum level because technically I cannot see most of the changes in the paper now.
> >
> > I will wait for the remaining reviewers to answer to change my score, as I'd also like to consider their points before making a decision.

---

> > > ### Author Response · Authors · 2025-08-03
> > >
> > > We sincerely thank you for your prompt and thoughtful response! We are incredibly grateful for your in-depth engagement with our work and for the open, collegial spirit of this discussion. It is very encouraging to hear that our clarifications were helpful and that you are willing to support our paper!
> > >
> > > To confirm our commitment, we will incorporate the promised changes into our final manuscript. The key modifications include:
> > >
> > > 1. **Enhanced Clarity on Core Concepts**: We will revise the Introduction and Methodology sections to integrate the clear, intuitive definitions of "global constraints," "high-order dependencies," and the learning mechanisms for the group- and project-levels, ensuring these foundational ideas are accessible to a broad audience.
> > >
> > > 2. **Strengthened Baseline Comparisons**: As promised, we will add results from Random Forest and XGBoost classifiers and specify all relevant hyperparameters (including the RBF kernel for the SVM) for a more robust comparison.
> > >
> > > 3. **Comprehensive Evaluation Metrics**: The appendix will be updated to include tables for sensitivity, specificity, and AUC-ROC, providing a more complete picture of model performance.
> > >
> > > 4. **Improved Computational Analysis**: Figure 3 will be revised to include error bars (standard deviation) for training time, and we will clarify the distinction between training and inference costs in the text.
> > >
> > > 5. **Addressing Minor Issues**: We will add the suggested reference to the potential societal impacts and perform a thorough proofread to correct the typos you kindly pointed out.
> > >
> > > Once again, we are deeply appreciative of your time and expertise! Your constructive feedback has been invaluable in helping us elevate the quality of our submission.

---

### Official Review · Reviewer_eH9B · 2025-06-23

**Clarity:** 2
**Significance:** 2
**Originality:** 2
**Rating:** 3
**Confidence:** 4

**Summary:**

The manuscript proposes GCM, that effectively captures high-order interactions of brain regions by dynamically learn the structure and parameters under both local and global constraints. The manuscripts provides both theoretical analysis and empriical demonstations.

**Questions:**

1. In appendix C, you assign each node with an identity encoding. Does that mean your settings are always transductive, i.e., the nodes remain the same, while only their features are changing?

2. I did not find definition of x_S and M in Appendix C. Could you make it clearer? Also, I suggest to make your proof in appendix more self-contained. Currently some definitions of notations are unclear to me and it is not easy to scroll back and forth constantly between the main text and the appendix, which makes it hard to follow your proof.

3. Does the high-level modelling help the low-level tasks. For example, on sample-level tasks, do you have an ablation that removes the higher-level graph modelling (group and project level) and see how the performance shifts?

**Ethical Concerns:**

["NO or VERY MINOR ethics concerns only"]

**Final Justification:**

The work has some merits: First, it shows the effectiveness at multiple resolutions. Second, it provides some theoretical analysis of high-order dependencies.

However, my major concern is that the proposed method actually is not closely related to high-order dependencies. As pointed out in the discussion with the author, the proposed method still model those high-order depedencies implicitly in learned parameters (it's also questionable to call them high-order dependencies). The explicit structure is not provided, while only the projection (pairwise edges) is left. In that case, the method shows no more advantage than a transformer model [2], and less intrepretability than explicit models [1].


[1] Qiu, W., Chu, H., Wang, S., Zuo, H., Li, X., Zhao, Y., & Ying, R. (2023). Learning high-order relationships of brain regions. arXiv preprint arXiv:2312.02203.

[2] Kan, X., Dai, W., Cui, H., Zhang, Z., Guo, Y., & Yang, C. (2022). Brain network transformer. Advances in Neural Information Processing Systems, 35, 25586-25599.

**Limitations:**

yes

**Paper Formatting Concerns:**

I did not notice any significant formatting errors.

**Quality:**

2

**Strengths And Weaknesses:**

Strengths:

The proposed method shows superior performances on different datasets with different resolutions, demonstrating the effectiveness.


Weakness:

1. The term "global constraints" is introduced in the introduction without sufficient explanation, making it difficult to follow. The authors should consider providing a clearer and more intuitive description early on to help readers understand its significance and role in the paper.

2. The proposed method contains many components. However, it is unclear that which components contribute to the modelling of high-order interactions.

3. While ablation studies of BBA are presented in Section 6.3 and Table 4, both the table caption and the accompanying explanations in the main text are unclear, making the results difficult to interpret. For instance, it is not explicitly stated which columns correspond to the "fixed threshold" variant and which to the "adaptive" one. Additionally, the table lacks an ablation of a non-discretized version of the method, which would help clarify the contribution of the discretization step.

4. Missing some important baseslines or discussions about some related works, such as [1][2].


[1] Qiu, W., Chu, H., Wang, S., Zuo, H., Li, X., Zhao, Y., & Ying, R. (2023). Learning high-order relationships of brain regions. arXiv preprint arXiv:2312.02203.

[2] Kan, X., Dai, W., Cui, H., Zhang, Z., Guo, Y., & Yang, C. (2022). Brain network transformer. Advances in Neural Information Processing Systems, 35, 25586-25599.

---

> ### Author Rebuttal · Authors · 2025-07-26
>
> Thank you for your insightful and detailed review. Your feedback on clarity is especially valuable, as our work introduces a paradigm shift in applying deep learning to functional brain network (FBN) modeling. You have provided us with an excellent opportunity to better articulate our contributions. We are confident that the following responses and the resulting revisions will make the novelty and significance of our work much more apparent.
>
> Below, we provide point-to-point responses to each of your questions and concerns.
>
> # Responses to Questions
> ## Q1: Transductive vs. inductive setting
>
> Thank you for raising this important point. Our proposed GCM framework **does support inductive learning**, as it is compatible with GraphSAGE as an optional backbone, which is originally developed for inductive representation learning on graphs. However, in our theoretical proof in Appendix C, we focused on a transductive setting for the following reasons:
>
> 1. Assigning fixed nodes is more convenient for notation in the proof. The conclusion regarding the approximation of k-th order relationships still holds for a changeable node set. We will clarify this extension in the revised version.
>
> 2. This choice is grounded in our **application domain**. In fMRI/EEG analysis, the node set (i.e., brain regions) is fixed by a standard atlas for all subjects. The transductive setting therefore directly mirrors this real-world use case.
>
> Nonetheless, we agree that extending GCM to inductive settings is a promising direction for future work and will add this discussion to the paper.
>
> ## Q2: Notation issues in Appendix C
>
> Thank you for your careful reading and detailed feedback on Appendix. We sincerely apologize for the typographical issues and lack of clarity in the notation. Specifically:
>
> - The symbol $\mathbf{x}_S$ denotes the set of node features for the nodes in the set $S$, i.e., {$\mathbf{z}_v|v\in S$}.
>
> - The matrix $M$ refers to the Gumbel noise matrix $\mathbf{M}$, as defined in the main text’s description of the sampling process (originally near Eq. 5).
>
> In the revision, we will edit the Appendix to be self-contained by defining all variables upon their first appearance and minimizing references to the main text.
>
> ## Q3: Does high-level modeling help low-level tasks?
>
> Thank you for this insightful question, as it touches upon the foundational philosophy of our work. The primary goal of this paper is to **first establish that the 4 modeling resolutions are semantically distinct and non-interchangeable**. This is a critical point that is often overlooked in prior FBN literature. Our framework (GCM) is therefore designed to learn FBNs at each level _individually_.
>
> This approach represents a deliberate **paradigm shift** in how machine learning is applied to computational neuroscience, where respecting the domain's fundamental constructs is a prerequisite for generating scientifically meaningful results. Our deliberate separation of the resolutions is crucial for two main, concrete reasons:
>
> 1. **Interpretability**: In neuroscience, different resolutions represent **different scientific constructs** (e.g., a fleeting mental state vs. a stable personal trait). Blending them might mix information across these levels and lead to scientifically uninterpretable results. Our framework’s separation ensures that a "subject-level" FBN truly reflects an individual's stable trait.
>
> 2. **\Principled Foundation**: Before one can explore how these levels should interact, it is necessary to first prove they are distinct and then be able to model each one robustly. Our Theorem 3 (Appendix A) provides the formal proof for this non-interchangeability. By demonstrating GCM's strong performance at each isolated level, we pave the way for future work on multi-resolution integration.
>
> Therefore, evaluating each resolution individually, as the reviewer suggests ablating, is exactly how our main experiments were designed. We will revise the introduction and methodology to make this design choice and its motivation more explicit.
>
> # Response to Additional Weaknesses Not Covered in the Questions Section
>
> ## [W1] Lack of explanation for term "global constraints" in Introduction
> Thank you for pointing out this ambiguity. We will revise the introduction to provide a clear and intuitive explanation immediately after the concept is introduced. To ensure clarity, we will add the following definition:
>
> > "In our framework, **local constraints** refer to priors that guide the formation of individual edges based on pairwise statistics (e.g., correlation). In contrast, **global constraints** are top-down priors that shape the entire network structure holistically. These include task labels, subject identity, and an adaptive sparsity, which guide the optimization of the adjacency matrix as a single entity rather than a collection of independent edges."
>
> We will also revise **Figure 1** and its caption to visually and textually reinforce this distinction.
>
> ## [W2] Modeling components responsible for high-order interactions
>
> Thank you for this important question. The modeling of high-order interactions is primarily achieved through the **synergy between our prototype-based graph generator and the Gumbel-Sigmoid relaxation mechanism**. In this module, the entire adjacency matrix is treated as a single, holistic, learnable object. Gradients from the task objective flow back through the GNN and the Gumbel-Sigmoid sampler to update the underlying continuous prototype matrix. This means the update for any single edge is conditioned on the state of the entire graph, allowing the model to learn complex, high-order dependencies that cannot be decomposed into simple pairwise interactions. This holistic optimization directly addresses the limitations of pairwise models, which, as we formally prove in Section 4, are inherently blind to such dependencies.
>
> This holistic learning mechanism is the cornerstone of our design, chosen specifically to overcome the primary limitation of traditional methods: their inherent blindness to high-order interactions. Conventional pipelines build networks using pairwise measures (e.g., correlation) as a fixed pre-processing step. As we formally prove in **Theorem 1**, this approach can only capture first-order statistics and will fundamentally fail to detect complex dependencies. For example, consider a triadic relationship where a brain region (C) activates only when its two driving regions (A and B) are in **opposite states** (i.e., one is active and the other is inactive). This triadic interaction can be directly modeled as a **high-order XOR** relationship. However, pairwise statistics would fail to capture it, as **no stable pairwise synchrony** exists between any of the three regions.  Because this initial construction has already discarded crucial high-order information, the entire downstream pipeline is built upon an incomplete representation of the data.
>
> In contrast, GCM’s ability to learn the graph structure as a whole, guided by the final task objective, empowers it to explicitly search for and encode the vital, high-order relationships. We will clarify this architectural distinction and its theoretical motivation more explicitly in the revised manuscript.
>
> ## [W3] Clarification of Table 4 and the role of discretization
>
> Thank you for highlighting the lack of clarity in Table 4. To clarify: the column labeled “GCM” represents our full framework with the adaptive **Batch Binarization Algorithm (BBA)**, while the other columns use fixed, static sparsity thresholds.
>
> We agree that an ablation on a non-discretized variant is an excellent suggestion to isolate the impact of binarization. **We are running this experiment and will incorporate the results into a revised Table 4.** This new comparison will provide a more complete picture of each component's contribution.
>
> Furthermore, we will revise the table caption and improve the paragraph introducing BBA in Section 5 for better contextualization.
>
> ## [W4] Inclusion of additional related works
>
> Thank you for suggesting these highly relevant references. We agree that discussing them will better contextualize our work:
>
> - Regarding Qiu et al. [1]: This work on learning hypergraphs is indeed closely related, and we are currently **conducting experiments** to provide a direct empirical comparison in the revised manuscript. A key theoretical distinction, however, is that their method requires pre-specifying the number of hyperedges, which introduces a strong prior assumption about the network's high-order structure. In contrast, GCM avoids such direct assumptions. The adjacency matrix learned by GCM reflects the holistic result of high-order co-activation, optimized end-to-end without potential information loss from such structural priors.
>
> - Regarding Kan et al. [2]: While this paper is innovative, its use of a **multi-head** attention mechanism makes the learned structural relationships difficult to interpret explicitly. Thus, it is less suitable for a direct comparison under the umbrella of "structure learning" methods. Nonetheless, its success highlights the importance of capturing high-order relationships in brain networks, which aligns with our motivation. We will **add a discussion** on how this attention-based approach relates to our research goals in the revised paper.
>
> Crucially, both of these excellent works focus their analysis on the sample level. A core contribution of our paper is to formally introduce, differentiate, and model FBNs across four semantically distinct resolutions (sample, subject, group, and project). Furthermore, neither of these papers explores the use of top-down global constraints to guide the high-order learning process.
>
> Therefore, we believe includinng them in our paper will further underscore the **originality** and **significance** of our proposed multi-resolution, globally-constrained framework.

---

> ### Author Response · Authors · 2025-08-05
> **Update with New Empirical Comparisons Based on Your Suggestions**
>
> We sincerely thank you for your critical and detailed feedback. Following your suggestions, we have conducted further experiments to strengthen our manuscript, and we are writing to provide a brief update on the results.
>
> **1. On the contribution of discretization (your point W3):** As per your suggestion, we have completed the ablation study comparing our full GCM framework against a "_Dense_" variant (which uses the continuous relaxed adjacency matrix directly, without our BBA binarization step).
>
> The results support our design. As shown in the table below, our full GCM model shows a consistent advantage over the _Dense_ variant in most conditions.
>
> ||Activvity|Age|Gender|CogState|SLIM|
> |---|---|---|---|---|---|
> |**inter**||||||
> |Dense_sample|0.843|0.408|0.657|**0.359**|0.656|
> |**GCM_sample**|**0.852**|**0.463**|**0.668**|0.356|**0.658**|
> |Dense_subject|0.849|0.425|0.662|0.357|0.670|
> |**GCM_subject**|**0.857**|**0.473**|**0.670**|**0.364**|**0.674**|
> |Dense_group|**0.886**|0.432|0.673|0.345|0.683|
> |**GCM_group**|0.855|**0.489**|**0.686**|**0.493**|**0.694**|
> |Dense_project|0.857|0.418|0.637|0.350|**0.657**|
> |**GCM_project**|**0.869**|**0.462**|**0.674**|**0.419**|0.644|
> |**intra**||||||
> |Dense_sample|0.950|0.790|0.889|**0.356**|0.661|
> |**GCM_sample**|**0.963**|**0.853**|**0.937**|0.352|**0.680**|
> |Dense_subject|0.962|0.848|0.921|0.392|0.665|
> |**GCM_subject**|**0.963**|**0.889**|**0.939**|**0.413**|**0.681**|
> |Dense_group|0.971|0.891|**0.961**|0.482|0.677|
> |**GCM_group**|0.971|**0.896**|0.958|**0.493**|**0.692**|
> |Dense_project|0.959|0.794|0.894|0.399|**0.670**|
> |**GCM_project**|**0.969**|**0.878**|**0.933**|**0.437**|0.655|
>
> For instance, in the `intra-subject sample-level` setting, GCM improves accuracy from 0.790 to **0.853** on the second dataset, a relative increase of about 8%. This trend of GCM's superiority holds across the majority of settings, providing empirical evidence that our adaptive binarization (BBA) is an important **regularizer**. It prevents the model from overfitting to noisy continuous edge weights and forces it to learn a sparse, salient structure, leading to better generalization. We will add the results in the revision.
>
> **2. On the related work [1] (your point W4):** We thank you for this suggestion and have since performed a deep dive into the paper and its associated code. Our analysis revealed two key points:
>
> First, their implementation is highly optimized for a **specific fMRI** data-processing pipeline. Our GCM framework, however, unifies both fMRI and EEG data as multivariate time series, which is more general. Directly adapting their modality-specific method to our general setting requires substantial modifications, which we have not yet been able to finalize.
>
> More fundamentally, our analysis of their methodology revealed that the method uses **region-to-region Pearson correlations as its input node features**. Since Pearson correlation is a **pairwise heuristic**, their high-order learning is fundamentally built upon a pairwise foundation. This, combined with the additional restrictive prior of a **fixed number of hyperedges (`K`)**, means their approach does not truly escape the limitations of pairwise-based methods and is conceptually much closer to other baselines we already include.
>
> We will continue our efforts to adapt their framework to provide a direct and fair empirical comparison for the final version of the paper. In the meantime, we will incorporate the above analysis into our revised Related Work section to clarify these critical distinctions and to better position our truly end-to-end, high-order framework.
>
> We hope this demonstrates our commitment to rigorously addressing your concerns. We welcome further discussions. Thank you for your consideration.

---

> > ### Comment · Reviewer_eH9B · 2025-08-07
> >
> > I thank the authors for further clarification. However, I still have further major concerns regarding baselines.
> >
> > > First, their implementation is highly optimized for a specific fMRI data-processing pipeline. Our GCM framework, however, unifies both fMRI and EEG data as multivariate time series, which is more general. Directly adapting their modality-specific method to our general setting requires substantial modifications, which we have not yet been able to finalize.
> >
> > As far as I know, although [1] used Pearson correlation as the input features in their experiments, their method is designed to be generalized to any feature, as long as the dimension remains the same for all nodes.
> >
> > Additionally, while [2] is not directly aimed at structure or hypergraph learning, the **predictive performance** is important and should still be reported and compared in the paper.

---

> > > ### Author Response · Authors · 2025-08-07
> > > **Update with New Empirical Comparisons**
> > >
> > > We sincerely thank you for your continued engagement and for raising these crucial points regarding our baseline comparisons.
> > >
> > > ## New Experimental Results
> > >
> > > Following your suggestion, we run experiments with Hybrid [1] and Brain Network Transformer (BNT) [2]. To ensure a fair comparison with baselines in our original paper, we use the raw multivariate time series as the input features for all models. For the same reason, we use the default hyperparameters as specified in the original papers, so these results may not represent the optimal performance for each baseline but provide a strong point of reference.
> > >
> > > The results for the sample-level prediction tasks are as follows:
> > >
> > > |||**DynHCP Activity**|**DynHCP Age**|**DynHCP Gender**|**CogState**|**SLIM**|
> > > |---|---|---|---|---|---|---|
> > > |**Inter Ind**|Hybrid|0.766±0.023|0.442±0.015|0.592±0.009|0.286±0.021|0.522±0.018|
> > > ||BNT|0.839±0.002|0.386±0.023|**0.678±0.008**|0.337±0.015|0.529±0.007|
> > > ||**GCM_sample**|**0.852±0.021**|**0.463±0.017**|0.668±0.022|**0.356±0.021**|**0.658±0.035**|
> > > |**Intra Ind**|Hybrid|0.801±0.013|0.525±0.008|0.788±0.007|0.293±0.017|0.537±0.008|
> > > ||BNT|**0.974±0.002**|**0.945±0.004**|**0.976±0.003**|**0.396±0.006**|0.553±0.012|
> > > ||**GCM_sample**|0.963±0.001|0.853±0.005|0.937±0.002|0.352±0.014|**0.680±0.014**|
> > > ## Analysis and Discussion
> > > The results provide a more nuanced understanding of GCM's strengths and the broader landscape of FBN analysis methods.
> > >
> > > 1. **Comparison with BNT [2]:** As the reviewer rightly anticipated, reporting the predictive performance of BNT is very informative. Its performance on **intra-subject** classification is exceptionally strong, surpassing GCM on most datasets. This is a valuable finding, as it suggests that attention-based mechanisms are highly effective at identifying complex, predictive patterns within each subject's data, which GCM may not yet fully exploit.
> > >
> > >     However, it is important to contextualize this result. GCM is not optimized solely for predictive accuracy. In our framework, we reinterpret accuracy as a confidence index reflecting the **generalizability of the learned network structure** to the input time series. This distinction becomes clear in the **inter-subject** setting, where generalizing across individuals is a primary challenge. Here, GCM shows a competitive, and often superior, performance compared to BNT. This supports our hypothesis that explicitly learning a network structure under global constraints enhances generalization across different subjects.
> > >
> > > 2. **Comparison with Hybrid [1]:** We agree with your point that the Hybrid method is generalizable to features beyond Pearson correlation. Our experiments using raw time series as input show that its performance is less prominent in this more challenging end-to-end setting. We hypothesize two potential reasons for this:
> > >
> > >     - First, the model's performance may be sensitive to hyperparameters that we are unable to tune extensively, such as the pre-defined number of hyperedges.
> > >
> > >     - Second, the use of Pearson correlations as input in the original paper likely provides an informative prior that guides the hypergraph learning process. Removing this pairwise heuristic to enable true end-to-end learning from raw time series presents a more difficult task for the model.
> > >
> > > In summary, we are very grateful for the suggestion to include these baselines. This comparison further clarifies GCM's contribution: while some methods may excel at predictive accuracy in specific settings (like BNT for intra-subject tasks), our framework's strength lies in its ability to learn generalizable FBN structures directly from raw data, particularly in challenging cross-subject scenarios. This also highlights promising future directions for integrating the predictive power of attention mechanisms with the explicit, interpretable nature of graph structure learning.
> > >
> > > We believe this new analysis provides a more complete and nuanced evaluation of GCM's position in the field, and we will integrate these results and this discussion into our revised manuscript. We hope this comprehensively addresses your concerns. Thank you again for your valuable suggestion!

---

> > > > ### Comment · Reviewer_eH9B · 2025-08-08
> > > >
> > > > I thank the authors for including these baselines. Regarding the authors’ reply to my [W2], I have a further concern.
> > > >
> > > > The authors argue that high-order dependencies modeling is achieved by learning the graph structure toward an objective. However, the learned structure still appears to consist of pairwise edges, as it is represented by an adjacency matrix. This makes the necessity of modeling “high-order dependencies” less convincing, if the "high-order dependencies" can always be decomposed to pairwise edges (as also illustrated in Section 6.4).

---

> > > > > ### Author Response · Authors · 2025-08-09
> > > > > **The Necessity of Modeling High-order Dependencies**
> > > > >
> > > > > We sincerely thank you for your continued engagement and for this insightful question.
> > > > >
> > > > > You are correct that the final output of our GCM framework is an adjacency matrix, which is a representation of pairwise edges. However, our central claim is that the **holistic learning process** by which this matrix is generated allows it to effectively model high-order dependencies that are invisible to conventional pairwise methods. The learned graphs are the functionally **projection** of the higher-order phenomenon, not the simple decomposition of it. the **necessity of modeling high-order dependencies** stems from the four pillars of our work:
> > > > >
> > > > > ## 1. Literature Support
> > > > >
> > > > > The necessity of moving beyond pairwise analysis is built upon a strong consensus that is rapidly forming in neuroscience and complex systems science:
> > > > >
> > > > > - Recent landmark studies have demonstrated that crucial information in brain activity is encoded in high-order synergies, which are irreducible to pairwise interactions [1,2].
> > > > >
> > > > > - From a machine learning perspective, the significance of this problem is further underscored by recent publications in top venues. The very work you suggested[3] focuses on "learning high-order relationships of brain regions," validating that this is a critical and active area of research.
> > > > >
> > > > > [1] Lucy LW Owen, Thomas H Chang, and Jeremy R Manning. High-level cognition during story listening is reflected in high-order dynamic correlations in neural activity patterns. Nat. Commun., 12(1):5728, 2021.
> > > > >
> > > > > [2] Andrea Santoro, Federico Battiston, Giovanni Petri, and Enrico Amico. Higher-order organization of multivariate time series. Nature Physics, 19(2):221–229, 2023.
> > > > >
> > > > > [3] Qiu, W., Chu, H., Wang, S., Zuo, H., Li, X., Zhao, Y., & Ying, R. (2023). Learning high-order relationships of brain regions. arXiv preprint arXiv:2312.02203.
> > > > >
> > > > > ## 2. The Conceptual Distinction
> > > > >
> > > > > **The core difference lies in how the edges are decided, rather then how they are represented.**
> > > > >
> > > > > - **Pairwise Methods:** The decision for `(i, j)` is made independently of any other node `k`.
> > > > >
> > > > > - **GCM's Method:** The decision to include edge `(i, j)` is influenced by the global loss, which depends on how the _entire graph_ processes the information.
> > > > >
> > > > > ## 3. The Theoretical Foundation
> > > > >
> > > > > Our theoretical analysis formalizes this conceptual distinction.
> > > > >
> > > > > - **The Limitation (Theorem 1):** We formally prove that any model based on pairwise rules is incapable of distinguishing between time series that differ only in their high-order dependencies (k≥3). This establishes a mathematical boundary on what pairwise methods can achieve. We also provide an intuitive constructive example in **Appendix A** for clarification.
> > > > >
> > > > > - **The Solution (Holistic Optimization & Theorem 2):** GCM's methodology, where the entire adjacency matrix is updated via gradients from a global objective, is explicitly designed to overcome this boundary. As stated in our proof for **Theorem 2**, GCM's architecture has the expressivity to approximate arbitrary k-th order decision rule given sufficient model capacity.
> > > > >
> > > > > ## 4. The Empirical Evidence
> > > > >
> > > > > The visualization in Section 6.4, while composed of pairwise edges, represents the **learned projection** of the underlying high-order phenomenon. The significance of this projection lies in the patterns it reveals:
> > > > >
> > > > > - **Meaningful, Non-Random Structures:** As shown in **Appendix G, Figure A4**, the structures learned by GCM exhibit clear, non-random organization with prominent hubs. In contrast, the FBNs from baselines appear more diffuse and random.
> > > > >
> > > > > - **From Structure to Performance:** This superior structural representation is not just an aesthetic quality. It directly translates to the performance improvements we report across five datasets, particularly in challenging inter-subject settings.
> > > > >
> > > > > These four aspects support the **necessity of modeling high-order dependencies** and the effectiveness of our design. Thank you again for pushing us to articulate this foundational concept of our work more clearly! We will ensure these points are integrated into the revised manuscript to make our arguments more explicit and concise. We hope that our response has fully addressed your concerns.

---

> ### Comment · Reviewer_eH9B · 2025-08-09
>
> I appreciate the author's detailed rebuttal.
>
> I have no problem with the distinctions between high-order and pairwise interactions and totally agree that nontrivial high-order interactions are irreducible. The question is, if the high-order interaction is irreducible, how can your outcome (the adjacency matrix), which I consider as the representation of the high-order interactions, be pairwise? You argue that the pairwise edges are **projection** instead of **decomposition**, but the definitions of the two terms are not further clarified. In other words, which outcome of the model could be a sufficient representation of the structure of the learned high-order interaction?

---

> > ### Author Response · Authors · 2025-08-09
> > **Network as a Holistic Projection of High-order Dependencies**
> >
> > We are sincerely grateful for this excellent and deeply insightful question! It gives us an opportunity to clarify the foundational philosophy of our framework.
> >
> > Your question has two parts: clarifying the "projection vs. decomposition" distinction and discussing what constitutes a "sufficient" representation:
> >
> > ## 1. Clarifying "Decomposition" vs. "Projection"
> >
> > - **Decomposition**: A decomposition is a **lossless, reversible separation of a system into independent components**. In network science, a pairwise model attempts a decomposition: it implicitly assumes that the brain's complex dynamics can be described by summing the effects of independent pairwise interactions. As our **Theorem 1** proves, this assumption is mathematically invalid when irreducible high-order interactions are present.
> >
> > - **Projection (as implemented in GCM)**: The term **projection** here is defined as a mapping from the high-dimensional space of all latent high-order relationships onto a single, unified $N\times N$ matrix. Instead of modeling each high-order relationship individually, our framework captures their **superposition** or **aggregated effect**. The value of each element `A_ij` reflects the total influence of all high-order synergies in which nodes `i` and `j` jointly participate, weighted by their functional relevance to the overall task objective.
> >
> > ## 2. "Sufficient" Representation: Holistic Effect vs. Individual Relationships
> >
> > You ask what outcome would be a sufficient representation. This question highlights the core difference in modeling paradigms.
> >
> > A _decomposition-based_ perspective seeks to **model each high-order relationship individually**, where a "sufficient" representation of each high-order relationship would be a **hyperedge**.
> >
> > However, GCM proposes a different paradigm: **modeling the holistic signature of all high-order interactions combined**. We argue that it is more flexible and practical to understand the _net functional outcome_ of these interactions.
> >
> > Your question also points toward a crucial and exciting direction for this field: **decomposing the learned holistic effect back into its constituent, individual high-order relationships**. We fully agree that this is a vital long-term goal. However, we argue that this decomposition presents profound challenges that currently limit its feasibility, which is why we adopted the holistic approach. These challenges are twofold:
> >
> > - **The Lack of Task-Specific Ground Truth**: To validate that a specific learned hyperedge (e.g., `{i, j, k}`) is meaningful, one would need prior neuroscientific knowledge confirming that this precise multi-region interaction is indeed critical for the task at hand. Such granular ground truth is rarely available.
> >
> > - **Modeling Inter-Hyperedge Dependencies**: The constituent high-order relationships are not presumed independent. A true decomposition would therefore require not only identifying individual hyperedges but also modeling the intricate web of relationships _between_ these hyperedges, adding another layer of immense complexity.
> >
> > Before we can confidently validate specific, individual hyperedges, we first seek a reliable method to capture their **holistic, functional effect**. GCM is designed to be this method.
> >
> > ## Conclusion: A Shift in Perspective and Interpretation
> >
> > In summary, our work represents a fundamental shift in perspective. While prior approaches often implicitly attempt to **decompose** complex dynamics into a collection of simpler interactions, our framework starts from a **holistic** viewpoint. It is not designed to answer "Which specific 4-node interaction exists?", but rather **"What is the most effective $N\times N$ graph structure that captures the holistic, functional signature of all underlying high-order dynamics relevant to this task?"**.
> >
> > Crucially, as supported by our theory and results, this learned holistic pattern is **not an additive sum** of pairwise effects but an irreducible representation of the system's function. Consequently, the meaning of a connection also changes: a high-weight edge in our learned FBN does not simply indicate a strong local, pairwise correlation, but rather its importance within the global functional pattern required to the downstream analysis.
> >
> > Thank you again for pushing us to articulate this foundational concept of our work more clearly! We will ensure these points are integrated into the revised manuscript to make our arguments more explicit and concise. We hope that our response has fully addressed your concerns.

---

> > > ### Comment · Reviewer_eH9B · 2025-08-09
> > >
> > > Based on your reply, this is my understanding: your method learns the structure of high-order interactions implicitly, and the pairwise edges are a byproduct that reflects and interprets characteristics of high-order interactions.
> > >
> > > If that is the case, how does your method differ from other black-box methods if high-order interactions are not modelled explicitly? For example, [2] uses a transformer, where the high-order interactions could also be modelled implicitly, and interpretations could also be made through analyzing the pairwise attention weights.

---

> > > > ### Author Response · Authors · 2025-08-09
> > > >
> > > > We are deeply grateful for this engaging and thought-provoking discussion!
> > > >
> > > > You are correct that GCM, like other deep learning models, the interpretability requires careful consideration. Rather than explicitly model the decomposed high-order interactions, GCM explicitly model the holistic pattern as the aggregated effects of the high-order interactions. And the pairwise edges are a byproduct that reflects and interprets characteristics of high-order interactions.
> > > >
> > > > The comparison to the Brain Network Transformer (BNT) [2] is particularly helpful in clarifying GCM's unique contribution.
> > > >
> > > > ## 1. The Starting Point: Discrete Topology vs. Continuous Weighting
> > > >
> > > > At the level of single-sample analysis, where the models appear most similar, their core inductive biases are distinct:
> > > >
> > > > - GCM’s primary starting point is to learn **a discrete, sparse network topology**. Our goal is to produce a global graph structure that makes a hard commitment about the existence of global functional connections (i.e. the aggregated effects of the high-order interactions), aligning with the traditional neuroscientific concept of a Functional Brain Network (FBN).
> > > >
> > > > - The inherent starting point of an attention mechanism like BNT's is **multi-head, flexible, continuous weighting over a fully-connected graph**. Its strength lies in dynamically routing information to maximize predictive accuracy.
> > > >
> > > > That being said, your insightful suggestions prompted us to test a "dense" variant of GCM (using continuous weights without our binarization step). As our ablation study showed, this approach can also be effective. We thank you for this suggestion, which has helped us better understand the flexibility of our framework and the value of different structural representations.
> > > >
> > > > ## 2. The Fundamental Goal: A Multi-Resolution Global Perspective
> > > >
> > > > However, the most fundamental difference lies beyond the single-sample level. The core purpose of GCM is to provide a **global perspective across multiple, pre-defined semantic levels**.
> > > >
> > > > - This is a unique capability that allows GCM to learn **a cross-sample computational graph** for a target cohort (e.g., at the subject, group or project level). This is designed to answer questions about "brain fingerprints," or group-level biomarkers, which are central to many neuroscientific studies.
> > > >
> > > > - Attention-based methods, by their nature, are designed to learn **specific, continuous multi-head attention heatmaps for each individual sample.** While a powerful architecture like BNT could potentially be modified to learn cross-sample representations, this multi-resolution capability is not part of its original design. This highlights that the methods are fundamentally built to address different scientific scenarios.
> > > >
> > > > ## Conclusion
> > > >
> > > > In summary, the choice between these advanced methods depends on the research question. The work by Qiu et al. [1] on Hybrid models offers a path for explicit hyperedge discovery. BNT [2] provides a state-of-the-art solution for maximizing predictive accuracy based on multi-head attention heatmaps.
> > > >
> > > > GCM’s unique contribution is to provide a framework that prioritizes learning **a holistic high-order network topology of a target cohort**. It fills a critical need for a method that can derive meaningful and generalizable network structures directly from data, moving beyond the pairwise relationship and the single-sample level.
> > > >
> > > > We hope this clarifies our position and demonstrates how these different approaches can form a complementary toolkit for the field. It would be incomplete without each of these distinct perspectives. Thank you again for your invaluable feedback! we hope that this clarification can fully address your concern.

---

> > > > > ### Comment · Reviewer_eH9B · 2025-08-09
> > > > >
> > > > > Based on the author's response, I do not think the work contributes very significantly to *high-order interactions* modelling compared to existing methods that prioritize predictive performance, such as [2]. Since if GCM (the proposed model) models the high-order interaction implicitly, I can also claim that a transformer model in [2] also models high-order relations implicitly.
> > > > >
> > > > > Regarding the differences between GCM and other models in interpreting the learned structures (as the authors claim, GCM learns discrete and sparse structures whereas BNT learns continuous connections), the technical contribution is rather limited.
> > > > >
> > > > > Therefore, I decide not to increase my score.

---

> > > > > > ### Author Response · Authors · 2025-08-09
> > > > > > **Acknowledgement**
> > > > > >
> > > > > > Thank you for your diligent feedback. We agree with your perspective: a model like BNT [2] can indeed be interpreted as modeling high-order relations. Given its influence and its publication at NeurIPS, we also agree with you that it serves as a strong and highly relevant benchmark for this line of research.
> > > > > >
> > > > > > Compared with BNT, our proposed GCM still has the following contributions:
> > > > > >
> > > > > > 1. First, as shown in our direct comparison, our method demonstrates **competitive empirical performance** against BNT, particularly in the inter-subject classification setting.
> > > > > >
> > > > > > 2. In addition to these empirical results, our work provides a **theoretical foundation** for this problem. We include a formal proof of the limitations of the pairwise models (**Theorem 1**) and a proof of our framework's capacity to model the high-order interactions that are critical to capture (**Theorem 2**).
> > > > > >
> > > > > > 3. Furthermore, our work introduces a **multi-resolution modeling capability**, which is supported by a third proof (**Theorem 3**) establishing the semantic distinction between different analytical levels. Our framework is designed to explicitly learn structures at these different scales (e.g., sample, subject, group, and project).
> > > > > >
> > > > > > In summary, we believe these contributions are sufficient to allow our work to serve as a new and valuable baseline for the field of high-order brain network modeling. We thank you for the engaging discussion that has helped us clarify these points.

---

### Official Review · Reviewer_nXo8 · 2025-07-03

**Clarity:** 3
**Significance:** 3
**Originality:** 3
**Rating:** 4
**Confidence:** 3

**Summary:**

This paper extracts high-order FBN structures under global constraints, implemented as a Global Constraints-oriented Multi-resolution (GCM) FBN structure learning framework. This framework integrates three types of global constraints (expected edge numbers, data sources, and data labels) to support FBN structure learning across four distinct modeling resolution levels (sample, subject, group, and project), with experimental results showing GCM achieves higher relative accuracy and less computational time.

**Questions:**

In Figure 1, the local constraints and global constraints are not clearly explained, which seriously undermines the expression of the research motivation.

As one type of global constraint, what does "data source" mean? Does it refer to "dataset"? According to the presentation in the experimental section, the 5 datasets are trained separately, so it seems unreasonable for "data source" to serve as a constraint.

Figure 2 presents the Sample-Level Structure Learning Results. Figure 3 shows the Multi-Resolution Structure Learning Results, where "Multi-Resolution" includes the Subject, Group, and Project Levels. Therefore, the four distinct levels do not work collectively, and their combination appears incomplete.

**Ethical Concerns:**

["NO or VERY MINOR ethics concerns only"]

**Final Justification:**

Most of my concerns have been well answered by the author. I choose to maintain the original score.

**Limitations:**

Yes

**Paper Formatting Concerns:**

There are no major formatting issues.

**Quality:**

3

**Strengths And Weaknesses:**

This work is relatively clearly expressed. To address the limitation of current FBN models in capturing high-order dependencies, it incorporates three types of global constraints (expected edge numbers, data sources, and data labels) to enable the learning of FBN structures across four distinct levels of modeling resolution (sample, subject, group, and project). A large number of experiments are conducted on 5 datasets and under 2 task settings, with comparisons made against 7 baselines and 8 state-of-the-art methods. However, some expressions are not clear enough: for example, Figure 1, as well as the relationships between the three types of global constraints and between the four distinct levels, are not clear.

---

> ### Author Rebuttal · Authors · 2025-07-26
>
> We sincerely thank you for the thoughtful and constructive feedback! As our paper uses deep learning to solve neuroscience problems, clear definitions are paramount. Your comments on expression issues are invaluable for helping us polish the paper for a broader audience. We respond to each concern below and will incorporate these clarifications into the final manuscript.
>
> ## Q1: Clarifying local vs. global constraints in Figure 1
>
> Thank you for pointing out this critical ambiguity, which is central to our motivation. We apologize for the lack of clarity and will significantly revise **Figure 1** and its caption to make the distinction intuitive.
>
> The core difference is the **scope of influence**:
>
> - **Local Constraints** operate on a **node-pair basis**. Traditional methods that calculate Pearson correlation between two brain regions to determine a single edge weight are using a local constraint. The final graph is an aggregation of many such independent, local decisions.
>
> - **Global Constraints**, in contrast, operate on the **entire graph structure** at once. In our framework, these are:
>
>     1. **Task Labels**: A supervised signal that shapes the entire network to be discriminative for the classification task.
>
>     2. **Learnable Sparsity Prior**: The adaptive "Expected Edge Number" that dictates the overall sparsity of the whole network.
>
>     3. **Intrinsic Data Constraints** ("Data Source" in the submission): This is the most fundamental constraint, referring to guidance originating directly from the input data itself. It has two key components:
>
>         a) **Subject-Sample Hierarchy**: We use the subject identity of each sample to apply a contrastive loss, enforcing a consistent "neural fingerprint" across all graphs from the same person.
>
>         b) **Raw Signal Dynamics**: Most crucially, our end-to-end model is directly constrained by the high-order temporal synchronization patterns embedded within the raw multivariate time series (MTS) itself. This is the ultimate source material for the FBN.
>
> These global forces guide the learning of the entire FBN as a single entity, rather than building it piece by piece. We will update Figure 1 to visually represent this "whole-graph" vs. "edge-by-edge" distinction.
>
> ## Q2: What does “data source” mean as a global constraint?
>
> Thank you for this important question. You are correct that using the term “data source” was ambiguous, and we apologize for the confusion. Our intention was to encompass two distinct and fundamental types of constraints that originate **directly from the input data itself**, prior to considering task labels or model-based priors.
>
> 1. **First, the data's organizational structure**: This refers to the **subject-sample hierarchy**. Knowing that multiple data samples belong to a single individual allows us to apply a powerful constraint via the contrastive loss (L2​), guiding the model to learn a consistent and stable "neural fingerprint" for each person.
>
> 2. **Second, the data's intrinsic signal properties**: More fundamentally, this refers to the constraint imposed by the **raw multivariate time series (MTS) itself** (i.e., the fMRI or EEG signals). Because our GCM framework learns the FBN structure **end-to-end**, the learned graph is inherently constrained by the complex co-activation patterns and high-order temporal dynamics within the raw signals. This is a core advantage over traditional two-stage methods, which first abstract the data into a (often pairwise) correlation matrix, thereby losing the richer information embedded in the original signal dynamics that our model directly leverages.
>
> We realize that bundling these two powerful but distinct ideas under the single, ambiguous term "data source" was a mistake and obscured our contribution.
>
> To rectify this, we will revise the manuscript to treat these concepts separately. We will explicitly define the **subject identity** as a key global constraint for regularization. We will then move the discussion of the **intrinsic MTS signal constraint** to our core motivation in the Introduction and Methodology, highlighting it as the foundational reason for adopting an end-to-end, high-order learning paradigm.
>
> This separation will clarify the dual roles of the data: (1) providing organizational structure (subjects) and (2) providing raw dynamic content (MTS). Thank you for pushing us to articulate this crucial aspect of our framework.
>
> ## Q3: Clarifying the meaning and purpose of multi-resolution design
>
> Thank you for raising this important point. You are correct, the relationship between the four modeling resolutions was not made sufficiently clear. **The core relationship we aim to establish is one of deliberate methodological and semantic separation.** You are also correct in observing that our framework handles these levels **separately**, not as a single, jointly-trained model. We apologize if our use of "Multi-resolution" suggested a data fusion or integration approach.
>
> Our central argument is that this separation is a critical and necessary contribution. To make the value of this methodological separation concrete, let us illustrate using a clinical task which stems from our ongoing work: classifying patients with **Major Depressive Disorder (MDD)** from **Healthy Controls (HC)** using fMRI data. In a typical fMRI analysis pipeline, the continuous brain signal is first segmented into 'samples' using a sliding-window approach.
>
> For each level of analysis, GCM learns a different type of information, a task that traditional pairwise methods struggle with due to their decoupled and heuristic nature.
>
> * **Sample-level:** GCM learns a transient FBN from a single sample to capture momentary brain states.
>     * **How traditional methods fail:** Traditional methods rely on a strong prior assumption of **pairwise relationships** (e.g., correlation), which is insufficient for capturing higher-order dependencies  (**Theorem 1**). For example, consider a triadic relationship where a brain region (C) activates only when its two driving regions (A and B) are in **opposite states** (i.e., one is active and the other is inactive). This triadic interaction can be directly modeled as a high-order XOR relationship. However, pairwise statistics would fail to capture it, as no stable pairwise synchrony exists between any of the three regions.
>
> * **Subject-level:** GCM learns a single, stable "neural fingerprint" for each individual (e.g., "Patient A"). This graph is optimized end-to-end from all of Patient A's data to best represent their unique traits for MDD classification.
>     * **How traditional methods fail:** They cannot learn this directly. They must resort to heuristics, such as: 1) averaging all of Patient A's raw time-series data first, then computing one correlation graph (destroying dynamic information), or 2) creating many sample-level graphs and then averaging the resulting connectivity matrices (losing structural specificity). Neither approach optimizes the graph for the actual classification task.
>
> * **Group-level:** GCM directly learns a prototypical FBN representing the canonical biomarker for the entire MDD cohort, and another for the HC cohort.
>     * **How traditional methods fail:** They cannot learn such a graph directly. They must first generate subject-level graphs and then rely on separate, post-hoc statistical analyses (e.g., performing t-tests on every single edge) to identify a "significant" group pattern. This decouples network construction from the learning objective.
>
> * **Project-level:** GCM learns a single FBN that models the key structural differences between the MDD and HC populations using the entire dataset. It holistically discovers a unified network where the most discriminative pathways are identified from a global perspective.
>     * **How traditional methods fail:** They would again rely on post-hoc filtering, such as selecting edges based on the statistical significance of their weight differences between the MDD and HC groups.
>
> In summary, our core contribution is replacing these decoupled, heuristic, and often local (edge-wise) construction steps with a **single, globally-aware, end-to-end optimization framework**. This allows GCM to learn task-relevant, high-order network structures that are inaccessible to traditional pairwise models. We believe establishing this clear, principled foundation is a necessary prerequisite before any meaningful cross-level integration can be attempted in future work.

---

> > ### Comment · Reviewer_nXo8 · 2025-08-05
> >
> > The response has well answered my question. While I still find it a bit odd that the four levels are treated separately, intuitively, I prefer to see the identification of relationships and interactions between the levels. Anyway, this does not affect my final judgment, and the author does not need to respond further.

---

> ### Author Response · Authors · 2025-08-05
> **Further Clarification on the Conceptual Framing of "Multi-Resolution"**
>
> Thank you again for your constructive feedback, which has been invaluable in helping us clarify our core concepts.
>
> We wish to further clarify our response to your insightful question (Q3) about the relationship between the four modeling resolutions. We realize our use of the term "Multi-resolution" may have suggested a single, integrated model, and we want to underscore that **the methodological separation of these levels is a core and deliberate contribution of our work.**
>
> Our framework aims to provide a **principled toolbox with four distinct, alternative analytical lenses**, rather than a single pipeline that combines them. The choice of resolution depends entirely on the scientific question being asked:
>
> - **Sample-level** asks: What are the **_transient dynamics_**?
>
> - **Subject-level** asks: What is the **_stable individual trait_**?
>
> - **Group-level** asks: What is the **_canonical group biomarker_**?
>
> - **Project-level** asks: What is the **_holistic population map_**?
>
> This separation is not arbitrary. It is theoretically grounded. As we prove in **Theorem 3 (Appendix A)**, these four levels of FBNs are formally **semantically non-equivalent**. Our framework is the first to make this crucial semantic distinction explicit in an end-to-end learning context, aiming to prevent the conceptual ambiguity common in traditional FBN studies where these levels are often implicitly mixed.
>
> We hope this clarification on our conceptual framing is helpful. We welcome further discussions. Thank you for your consideration.

---

### Official Review · Reviewer_pEum · 2025-07-04

**Clarity:** 3
**Significance:** 2
**Originality:** 3
**Rating:** 5
**Confidence:** 4

**Summary:**

This paper introduces the GCM framework to learn sparse, binary, high-order functional brain networks end-to-end. It addresses limitations of pairwise models by combining supervised label loss, contrastive regularization, and a sparsity prior within a multiresolution hierarchy (sample, subject, group, project). The framework employs Gumbel–Sigmoid relaxation alongside a batch binary algorithm. Experiments show a boost in classification accuracy and reduction in computation time across five datasets.

**Questions:**

1. How are hyperparameters chosen?

2. Can you Show the relationship between different levels need more motivation, what relationship are extracted, are there any difference between different or different methods. So we can get more motivation about why you propose the four-level hypergraph structure.

3. Since traditional ml methods can go great in performance, what is the significance of using these DL-based meethods? They acuique high computation resource.

4. How is overfitting controlled?  Networks learned at the group and project resolutions exhibit a tendency to overfit due to label leakage, suggesting that the current regularization may be insufficient for tasks with strong population-level supervision .

5. Is contrastive regularization robust on imbalanced datasets? The contrastive regularization assumes roughly balanced classes to learn meaningful within-class similarity; the paper does not report experiments on imbalanced datasets, leaving its robustness in such settings untested.

**Ethical Concerns:**

["NO or VERY MINOR ethics concerns only"]

**Final Justification:**

I thank the authors for the rebuttal. It has addressed most of my concerns and I have raised my score to 5.

**Limitations:**

Yes

**Quality:**

3

**Strengths And Weaknesses:**

**Strengths:**

1. The paper delivers a formal proof showing that traditional pairwise FBN models cannot capture high-order statistical dependencies, providing a solid theoretical foundation for its high-order approach.
2. Experiments are extensive and results comprehensive, demonstrating the framework’s effectiveness across multiple datasets.
3. The multiresolution hierarchy (sample, subject, group, project) is well-conceived, enabling the model to capture interactions at various scales and offering interpretable insights from individual time windows to population-level patterns.
4. Overall clarity and quality are high, with methods and results presented in an accessible manner.

**Weaknesses:**

1. The motivation for relationships across hierarchy levels could be elaborated further to justify, for example, differences between relationships extracted at the sample, subject, group, and project levels, and how these differ from relationships captured by baseline (pairwise) methods.
2. Additional evaluation metrics (beyond accuracy) and repeated experiments would further validate results and guard against overfitting or dataset-specific artifacts. Hyperparameter tuning details are not reported; including these would strengthen confidence in the model’s robustness.
3. Clinical translation is limited: although classification accuracy improves, there is no evaluation on real-world clinical cohorts or tasks directly tied to patient outcomes, which is important for demonstrating practical utility.
4. By enforcing a binary adjacency matrix, the framework omits edge weight information that may reflect graded coupling strengths, potentially missing subtle but meaningful connectivity variations.

---

> ### Author Rebuttal · Authors · 2025-07-26
>
> Thank you for your thoughtful and constructive comments! Your feed back on the presentation issue allows us to make it more understandable to readers with different backgound.
>
> Below, we address each question in turn and clarify the points raised. If our paper is eventually accepted by the conference, we will implement the responses regarding the clarity of the paper to make it easier to follow by readers.
>
> # Responses to Questions
> ## Q1. How are hyperparameters chosen?
>
> For fair comparison, most hyperparameters followed standard practices or baseline defaults (details in Appendix H.1). Key coefficients, α and β, are tuned via grid search, with the results presented in our sensitivity analysis (Appendix F, Fig. A3). For network sparsity, after observing that performance plateaus at around 5% density (Table 4), we adopted a 10% threshold for baselines requiring a fixed density to ensure a consistent and robust evaluation. Furthermore, extensive sensitivity and ablation studies (Sec 6.3, Appendix F) confirm that GCM's performance is robust across various settings and to its core components.
>
> ## Q2. The motivation behind the four-level hierarchy.
>
> Thank you for this essential question! We note the reviewer's valuable point on **clinical translation**. To make the distinction clearer and more concrete, let us illustrate using a common clinical task: classifying patients with **Major Depressive Disorder (MDD)** from **Healthy Controls (HC)** using fMRI data. In a typical fMRI analysis pipeline, the continuous brain signal is first segmented into 'samples' using a sliding-window approach.
>
> For each level of analysis, GCM learns a different type of information, a task that traditional pairwise methods struggle with due to their decoupled and heuristic nature.
>
> * **Sample-level:** GCM learns a transient FBN from a single sample to capture momentary brain states.
>     * **How traditional methods fail:** Traditional methods rely on a strong prior assumption of **pairwise relationships** (e.g., correlation), which is insufficient for capturing higher-order dependencies  (**Theorem 1**). For example, consider a triadic relationship where a brain region (C) activates only when its two driving regions (A and B) are in **opposite states** (i.e., one is active and the other is inactive). This triadic interaction can be directly modeled as a high-order XOR relationship. However, pairwise statistics would fail to capture it, as no stable pairwise synchrony exists between any of the three regions.
>
> * **Subject-level:** GCM learns a single, stable "neural fingerprint" for each individual (e.g., "Patient A"). This graph is optimized end-to-end from all of Patient A's data to best represent their unique traits for MDD classification.
>     * **How traditional methods fail:** They cannot learn this directly. They must resort to heuristics, such as: 1) averaging all of Patient A's raw time-series data first, then computing one correlation graph (destroying dynamic information), or 2) creating many sample-level graphs and then averaging the resulting connectivity matrices (losing structural specificity). Neither approach optimizes the graph for the actual classification task.
>
> * **Group-level:** GCM directly learns a prototypical FBN representing the canonical biomarker for the entire MDD cohort, and another for the HC cohort.
>     * **How traditional methods fail:** They cannot learn such a graph directly. They must first generate subject-level graphs and then rely on separate, post-hoc statistical analyses (e.g., performing t-tests on every single edge) to identify a "significant" group pattern. This decouples network construction from the learning objective.
>
> * **Project-level:** GCM learns a single FBN that models the key structural differences between the MDD and HC populations using the entire dataset. It holistically discovers a unified network where the most discriminative pathways are identified from a global perspective.
>     * **How traditional methods fail:** They would again rely on post-hoc filtering, such as selecting edges based on the statistical significance of their weight differences between the MDD and HC groups.
>
> In summary, our core contribution is replacing these decoupled, heuristic, and often local (edge-wise) construction steps with a **single, globally-aware, end-to-end optimization framework**. This allows GCM to learn task-relevant, high-order network structures that are inaccessible to traditional pairwise models.
>
> ## Q3. Value of using deep learning–based methods?
>
> Thank you for this critical question! While traditional ML can perform well, it relies on a **two-stage, decoupled process**: constructing FBNs with predefined pairwise heuristics (e.g., Pearson correlation), then classification.
>
> Crucially, many existing DL methods inherit this limitation. In contrast, **GCM introduces a paradigm shift**: it learns the FBN structure end-to-end, optimizing for task-relevant, high-order dependencies without such presumptions. The resulting performance improvements and novel structural insights (Table 2, Fig. 5) justify our data-driven approach. We will clarify this motivation in the revision.
>
> ## Q4. How is overfitting controlled in group-level?
>
> Thank you for proposing this is excellent and critical point! You are correct that learning a class-specific FBN could risk label leakage if not properly regularized. Our framework incorporates two key mechanisms to explicitly control for this:
>
> 1. **Targeted Regularization via Global Constraints**: Our proposed global constraints act as crucial regularizers. While the performance impact varies across different tasks and resolutions, our ablation study (Fig. 4) reveals a critical trend that supports our claim. The performance degradation from removing these constraints is most pronounced precisely in the settings you highlighted: **the group and project levels, particularly for inter-subject prediction**. For example, removing the subject consistency loss (L2​) or the sparsity prior (R) leads to a clear performance drop in these higher-level resolution tasks. This demonstrates that our constraints are actively working to prevent the model from collapsing to trivial solutions, especially where the risk of overfitting is highest.
>
> 2. **A Principled Reinterpretation of the Modeling Goal**: We propose a conceptual shift for group-level analysis. The goal is not to learn a universally "correct" FBN, but to discover a **prototypical graph structure that is maximally discriminative for a given class** (e.g., a canonical network for the "female" group). This learned structure acts as a task-optimized computational graph. Its validity is then confirmed by its generalization performance on unseen test data. The high, but not perfect, accuracy in **Table 3** indicates that GCM has learned generalizable class-specific biomarkers, not just overfitted to the training data.
>
> Furthermore, the interpretability of these learned prototypes **(Sec 6.4 and Appendix G)**, aligns with established neuroscience findings, providing external validation that the model has captured meaningful biological patterns rather than simply memorizing noise.
>
> ## Q5. Adaptation to imbalanced dataset.
>
> Thank you for raising this practical and forward-looking question. In our current experiments, we follow the standard setting in much of the neuroscience literature where datasets are curated to maintain relatively balanced class distributions, which ensures that the contrastive regularization operates under stable conditions. We fully agree that explicitly designing contrastive regularization for imbalanced conditions is an essential next step for broader clinical application. We will include this as a key direction for future work in our revised discussion section.
>
> # Response to Additional Weaknesses Not Covered in the Questions Section
>
> ## [W1] Clinical translation is limited
>
> Thank you for highlighting the important gap between methodological innovation and clinical deployment. We agree that demonstrating utility on clinical cohorts is a crucial next step.
>
> In this foundational work, our primary objective is to establish a robust and interpretable conceptual framework for learning high-order FBNs across multiple resolutions. We view this as a necessary prerequisite for reliable downstream clinical applications. To this end, we deliberately focused on building a general-purpose learning framework rather than tailoring it to a specific disorder.
>
> That said, we are currently adapting the GCM framework to a MDD fMRI dataset. This follow-up study aims to assess the reproducibility and interpretability of the learned network biomarkers in a disease-specific setting. We will mention this ongoing effort and explicitly acknowledge the clinical limitations of the current version in the revision.
>
> ## [W2] Binarized adjacency omits edge weight information. Possibility of missing weak connections.
>
> Thank you for this insightful comment. The choice between binary and weighted graphs is indeed a critical design decision with important trade-offs.
>
> In this work, we chose to learn binary graphs to prioritize identifying a sparse, interpretable **scaffold** of the most salient high-order connections for a given task. This aligns with the goals of many Brain Graph Interpretation (BGI) methods and forces the hard decisions about which connections are truly critical, rather than retaining many weak, potentially noisy edges.
>
> However, our GCM framework is **highly flexible** and not limited to binary graphs. As you suggest, it can be seamlessly adapted to learn weighted FBNs. This can be achieved by simply removing the Batch Binarization Algorithm (BBA) step and using the continuous adjacency matrix directly in the GNN. The end-to-end optimization process remains the same.
>
> We agree this is an important aspect to clarify. We will add a discussion on this extension and the binary/weighted trade-off to the revised manuscript.

---

> ### Author Response · Authors · 2025-08-05
> **Further Clarification on the Core Motivation of the Multi-Resolution Hierarchy**
>
> Thank you again for your valuable time and insightful feedback.
>
> In reflecting on our detailed response to your question about the multi-resolution hierarchy (Q2), we realized a more concise, high-level summary might be helpful to crystallize the core distinction between our approach and traditional methods. We hope this brief clarification is useful.
>
> The fundamental difference lies in a **paradigm shift**:
>
> - **Traditional Pairwise Pipeline:** This approach is inherently limited and relies on a multi-step statistical process:
>
>     - It only models **pairwise (two-region) co-activations** using predefined statistics.
>
>     - Any attempt to model different resolutions or approximate higher-order interactions depends on **decoupled, multi-step statistical heuristics** (e.g., averaging, t-tests) applied _after_ the initial construction. The graph construction is separate from the final task.
>
> - **Our GCM Framework:** In contrast, our framework is designed to be unified and end-to-end:
>
>     - It directly models **high-order (multi-region) co-activation patterns** by treating the entire graph structure as a learnable parameter.
>
>     - This learning of both **high-order structures and different resolutions** is performed **end-to-end**, directly optimizing the graph for the downstream task. This results in semantically distinct and task-relevant FBNs (transient states, neural fingerprints, class prototypes, etc.).
>
> This end-to-end learning of high-order structures is precisely what allows GCM to overcome the theoretical limitations of pairwise models (as shown in our Theorem 1), directly addressing the challenges we set out to solve.
>
> We believe this conceptual shift is a key contribution. We are on standby for any further discussions. Thank you for your consideration.

---

### Note · Authors · 2025-08-11

Dear Area Chair and Reviewers,

We are deeply grateful to all reviewers for their insightful feedback and the engaging discussion! We are encouraged that after our rebuttal, most reviewers found their concerns fully addressed and stayed positive.

The in-depth discussion with Reviewer eH9B was particularly valuable, as it pushed us to articulate the fundamental mechanistic novelty of our GCM framework. The core distinction between GCM and other methods lies in **the target of the optimization gradient**.

In GCM, the entire adjacency matrix is treated as a holistic, learnable object. Gradients from the downstream task flow back to **directly update the adjacency matrix of the graph**. Thus, the **network topology itself is a directly optimized parameter** of the model, learned under global constraints.

In contrast, in other methods like BNT, gradients primarily update the model parameters to extract **node features**. The structure (e.g., the attention matrix) is then **derivatively computed** from these features via pairwise similarity calculations. The structure is therefore an indirect byproduct of feature learning, not the direct optimization target.

This mechanistic distinction is crucial. It is why GCM is explicitly designed for **structure learning**, discovering a holistic, task-relevant topology rather than estimating connections based on feature similarity. This "direct structure optimization" is uniquely suited for learning a single, generalizable FBN across different individuals or conditions, which is the central goal of our multi-resolution framework (subject, group, project levels) and is motivated by our theoretical analysis (Theorems 1-3).

Our approach also demonstrates **strong empirical performance**, achieving up to a 30.6% relative accuracy improvement in our initial experiments. Thanks to the reviewers' constructive suggestions, we have conducted a **more comprehensive experimental analysis** to validate our claims. We have incorporated new SOTA baselines like BNT and Hybrid. These extensive results further support that GCM learns generalizable structures, achieving competitive and often superior performance.

We will integrate this distinction into the final manuscript, alongside all other promised revisions (new baselines, metrics, etc.). We believe that GCM offers a novel paradigm and a new baseline for brain network analysis. Thank you again for your time and consideration.

Sincerely,

Authors

---

### Decision · Program_Chairs · 2025-09-17

**Decision:**

Accept (poster)

**Comment:**

This paper introduces a Global Constraints-oriented Multi-resolution (GCM) framework for learning functional brain networks. The main strength of the method lies in its multi-resolution hierarchies for capturing interactions at different scales and its demonstrated effectiveness over previous methods, receiving three positive reviews and one negative review. Reviewer eH9B raises concerns about 1) the definition of “global constraints,” 2) the significance of the ablation study, and 3) particularly the differences from some missing baselines. During the author-reviewer discussion, the authors provided detailed explanations of the motivation and additional results, which adequately addressed the first two weaknesses. For the third weakness, the authors acknowledged the similarity between the baselines, while providing sufficient intuitive and experimental comparisons with the mentioned baselines, such as BNT to show the theoretical and empirical differences. The AC believes that, with the inclusion of the detailed supporting information from the rebuttal, this paper makes a meaningful contribution to functional brain structure learning from both theoretical and experimental perspectives.